# Revisiting Sliced Wasserstein on Images: From Vectorization to Convolution

**Khai Nguyen**
Department of Statistics and Data Sciences
The University of Texas at Austin
Austin, TX 78712
khainb@utexas.edu

**Nhat Ho**
Department of Statistics and Data Sciences
The University of Texas at Austin
Austin, TX 78712
minhnhat@utexas.edu

## Abstract

The conventional sliced Wasserstein is defined between two probability measures that have realizations as *vectors*. When comparing two probability measures over images, practitioners first need to vectorize images and then project them to one-dimensional space by using matrix multiplication between the sample matrix and the projection matrix. After that, the sliced Wasserstein is evaluated by averaging the two corresponding one-dimensional projected probability measures. However, this approach has two limitations. The first limitation is that the spatial structure of images is not captured efficiently by the vectorization step; therefore, the later slicing process becomes harder to gather the discrepancy information. The second limitation is memory inefficiency since each slicing direction is a vector that has the same dimension as the images. To address these limitations, we propose novel slicing methods for sliced Wasserstein between probability measures over images that are based on the convolution operators. We derive *convolution sliced Wasserstein* (CSW) and its variants via incorporating stride, dilation, and non-linear activation function into the convolution operators. We investigate the metricity of CSW as well as its sample complexity, its computational complexity, and its connection to conventional sliced Wasserstein distances. Finally, we demonstrate the favorable performance of CSW over the conventional sliced Wasserstein in comparing probability measures over images and in training deep generative modeling on images[1].

## 1 Introduction

Optimal transport and Wasserstein distance [59, 51] have become popular tools in machine learning and data science. For example, optimal transport has been utilized in generative modeling tasks to generate realistic images [2, 58], in domain adaptation applications to transfer knowledge from source to target domains [9, 3], in clustering applications to capture the heterogeneity of data [21], and in other applications [29, 62, 63]. Despite having appealing performance, Wasserstein distance has been known to suffer from high computational complexity, namely, its computational complexity is at the order of $\mathcal{O}(m^3 \log m)$ [49] when the probability measures have at most $m$ supports. In addition, Wasserstein distance also suffers from the curse of dimensionality, namely, its sample complexity is at the order of $\mathcal{O}(n^{-1/d})$ [15] where $n$ is the sample size. A popular line of work to improve the speed of computation and the sample complexity of the Wasserstein distance is by adding an entropic regularization term to the Wasserstein distance [10]. This variant is known as entropic regularized optimal transport (or equivalently entropic regularized Wasserstein). By using the entropic version, we can approximate the value of Wasserstein distance with the computational complexities

---

[1]Code for the paper is published at https://github.com/UT-Austin-Data-Science-Group/CSW.

36th Conference on Neural Information Processing Systems (NeurIPS 2022).

being at the order of $\mathcal{O}(m^2)$ [1, 35, 36, 34] (up to some polynomial orders of approximation errors). Furthermore, the sample complexity of the entropic version had also been shown to be at the order of $\mathcal{O}(n^{-1/2})$ [39], which indicates that it does not suffer from the curse of dimensionality.

Another useful line of work to improve both the computational and sample complexities of the Wasserstein distance is based on the closed-form solution of optimal transport in one dimension. A notable distance along this direction is sliced Wasserstein (SW) distance [6]. Due to the fast computational complexity $\mathcal{O}(m \log_2 m)$ and no curse of dimensionality $\mathcal{O}(n^{-1/2})$, the sliced Wasserstein has been applied successfully in several applications, such as generative modeling [61, 13, 25, 47], domain adaptation [31], and clustering [26]. The sliced Wasserstein is defined between two probability measures that have supports belonging to a vector space, e.g, $\mathbb{R}^d$. As defined in [6], the sliced Wasserstein is written as the expectation of one-dimensional Wasserstein distance between two projected measures over the uniform distribution on the unit sphere. Due to the intractability of the expectation, Monte Carlo samples from the uniform distribution over the unit sphere are used to approximate the sliced Wasserstein distance. The number of samples is often called the number of projections and it is denoted as $L$. On the computational side, the computation of sliced Wasserstein can be decomposed into two steps. In the first step, $L$ projecting directions are first sampled and then stacked as a matrix (the projection matrix). After that, the projection matrix is multiplied by the two data matrices resulting in two matrices that represent $L$ one-dimensional projected probability measures. In the second step, $L$ one-dimensional Wasserstein distances are computed between the two corresponding projected measures with the same projecting direction. Finally, the average of those distances is yielded as the value of the sliced Wasserstein.

Despite being applied widely in tasks that deal with probability measures over images [61, 13], the conventional formulation of sliced Wasserstein is not well-defined to the nature of images. In particular, an image is not a vector but is a tensor. Therefore, a probability measure over images should be defined over the space of tensors instead of vectors. The conventional formulation leads to an extra step in using the sliced Wasserstein on the domain of images which is vectorization. Namely, all images (supports of two probability measures) are transformed into vectors by a deterministic one-one mapping which is the "reshape" operator. This extra step does not keep the spatial structures of the supports, which are crucial information of images. Furthermore, the vectorization step also poses certain challenges to design efficient ways of projecting (slicing) samples to one dimension based on prior knowledge about the domain of samples. Finally, prior empirical investigations indicate that there are several slices in the conventional Wasserstein collapsing the two probability measures to the Dirac Delta at zero [13, 12, 24]. Therefore, these slices do not contribute to the overall discrepancy. These works suggest that the space of projecting directions in the conventional sliced Wasserstein (the unit hyper-sphere) is potentially not optimal, at least for images.

**Contribution.** To address these issues of the sliced Wasserstein over images, we propose to replace the conventional formulation of the sliced Wasserstein with a new formulation that is defined on the space of probability measures over tensors. Moreover, we also propose a novel slicing process by changing the conventional matrix multiplication to the convolution operators [16, 18]. In summary, our main contributions are two-fold:

1. We leverage the benefits of the convolution operators on images, including their efficient parameter sharing and memory saving as well as their superior performance in several tasks on images [28, 19], to introduce efficient slicing methods on sliced Wasserstein, named *convolution slicers*. With those slicers, we derive a novel variant of sliced Wasserstein, named *convolution sliced Wasserstein* (CSW). We investigate the metricity of CSW, its sample and computational complexities, and its connection to other variants of SW.

2. We illustrate the favorable performance of CSW in comparing probability measures over images. In particular, we show that CSW provides an almost identical discrepancy between MNIST's digits compared to that of the SW while having much less slicing memory. Furthermore, we compare SW and CSW in training deep generative models on standard benchmark image datasets, including CIFAR10, CelebA, STL10, and CelebA-HQ. By considering the quality of the trained models, training speed, and training memory of CSW and SW, we observe that CSW has more favorable performance than the vanilla SW.

**Organization.** The remainder of the paper is organized as follows. We first provide background about Wasserstein distance, the conventional slicing process in the sliced Wasserstein distance, and the convolution operator in Section 2. In Section 3, we propose the convolution slicing and the

convolution sliced Wasserstein, and analyze some of its theoretical properties. Section 4 contains the application of CSW to generative models, qualitative experimental results, and quantitative experimental results on standard benchmarks. We conclude the paper In Section 5. Finally, we defer the proofs of key results and extra materials in the Appendices.

**Notation.** For any $d \geq 2$, $\mathbb{S}^{d-1} := \{\theta \in \mathbb{R}^d \mid ||\theta||_2^2 = 1\}$ denotes the $d$ dimensional unit hyper-sphere in $\mathcal{L}_2$ norm, and $\mathcal{U}(\mathbb{S}^{d-1})$ is the uniform measure over $\mathbb{S}^{d-1}$. Moreover, $\delta$ denotes the Dirac delta function. For $p \geq 1$, $\mathcal{P}_p(\mathbb{R}^d)$ is the set of all probability measures on $\mathbb{R}^d$ that have finite $p$-moments. For $\mu, \nu \in \mathcal{P}_p(\mathbb{R}^d)$, $\Pi(\mu, \nu) := \{\pi \in \mathcal{P}_p(\mathbb{R}^d \times \mathbb{R}^d) \mid \int_{\mathbb{R}^d} \pi(x, y)dx = \nu, \int_{\mathbb{R}^d} \pi(x, y)dy = \mu\}$ is the set of transportation plans between $\mu$ and $\nu$. For $m \geq 1$, we denotes $\mu^{\otimes m}$ as the product measure which has the supports are the joint vector of $m$ random variables that follows $\mu$. For a vector $X \in \mathbb{R}^{dm}$, $X := (x_1, \ldots, x_m)$, $P_X$ denotes the empirical measures $\frac{1}{m} \sum_{i=1}^m \delta_{x_i}$. For any two sequences $a_n$ and $b_n$, the notation $a_n = \mathcal{O}(b_n)$ means that $a_n \leq C b_n$ for all $n \geq 1$ where $C$ is some universal constant.

## 2 Background

In this section, we first review the definitions of the Wasserstein distance, the conventional slicing, and the sliced Wasserstein distance, and discuss its limitation. We then review the convolution and the padding operators on images.

**Sliced Wasserstein:** For any $p \geq 1$ and dimension $d' \geq 1$, we first define the Wasserstein-$p$ distance [59, 50] between two probability measures $\mu \in \mathcal{P}_p(\mathbb{R}^{d'})$ and $\nu \in \mathcal{P}_p(\mathbb{R}^{d'})$, which is given by $W_p(\mu, \nu) := \left( \inf_{\pi \in \Pi(\mu, \nu)} \int_{\mathbb{R}^{d'} \times \mathbb{R}^{d'}} ||x - y||_p^p d\pi(x, y) \right)^{\frac{1}{p}}$. When $d' = 1$, the Wasserstein distance has a closed form which is $W_p(\mu, \nu) = (\int_0^1 |F_\mu^{-1}(z) - F_\nu^{-1}(z)|^p dz)^{1/p}$ where $F_\mu$ and $F_\nu$ are the cumulative distribution function (CDF) of $\mu$ and $\nu$ respectively.

Given this closed-form property of Wasserstein distance in one dimension, the sliced Wasserstein distance [6] between $\mu$ and $\nu$ had been introduced and admitted the following formulation: $SW_p^p(\mu, \nu) := \int_{\mathbb{S}^{d-1}} W_p^p(\theta \sharp \mu, \theta \sharp \nu)d\theta$, where $\theta \sharp \mu$ is the push-forward probability measure of $\mu$ through the function $T_\theta : \mathbb{R}^{d'} \to \mathbb{R}$ with $T_\theta(x) = \theta^\top x$. For each $\theta \in \mathbb{S}^{d'-1}$, $W_p^p(\theta \sharp \mu, \theta \sharp \nu)$ can be computed in linear time $\mathcal{O}(m \log_2 m)$ where $m$ is the number of supports of $\mu$ and $\nu$. However, the integration over the unit sphere in the sliced Wasserstein distance is intractable to compute. Therefore, Monte Carlo scheme is employed to approximate the integration, namely, $\theta_1, \ldots, \theta_L \sim \mathcal{U}(\mathbb{S}^{d'-1})$ are drawn uniformly from the unit sphere and the approximation of the sliced Wasserstein distance is given by: $\widehat{SW}_p^p(\mu, \nu) \approx \frac{1}{L} \sum_{i=1}^L W_p^p(\theta_i \sharp \mu, \theta_i \sharp \nu)$. In practice, $L$ should be chosen to be sufficiently large compared to the dimension $d'$, which can be undesirable.

**Sliced Wasserstein on Images:** Now, we focus on two probability measures over images: $\mu, \nu \in \mathcal{P}_p(\mathbb{R}^{c \times d \times d})$ for number of channels $c \geq 1$ and dimension $d \geq 1$. In this case, the sliced Wasserstein between $\mu$ and $\nu$ is defined as:

$$SW_p(\mu, \nu) = SW_p(\mathcal{R} \sharp \mu, \mathcal{R} \sharp \nu), \tag{1}$$

where $\mathcal{R} : \mathbb{R}^{c \times d \times d} \to \mathbb{R}^{cd^2}$ is a deterministic one-to-one "reshape" mapping.

**The slicing process:** The slicing of sliced Wasserstein distance on probability measures over images consists of two steps: vectorization and projection. Suppose that the probability measure $\mu \in \mathcal{P}(\mathbb{R}^{c \times d \times d})$ has $n$ supports. Then the supports of $\mu$ are transformed into vectors in $\mathbb{R}^{cd^2}$ and are stacked as a matrix of size $n \times cd^2$. A projection matrix of size $L \times cd^2$ is then sampled and has each column as a random vector following the uniform measure over the unit hyper-sphere. Finally, the multiplication of those two matrices returns $L$ projected probability measures of $n$ supports in one dimension. We illustrate this process in Figure 1.

**Limitation of the conventional slicing:** First of all, images contain spatial relations across channels and local information. Therefore, transforming images into vectors makes it challenging to obtain that information. Second, vectorization leads to the usage of projecting directions from the unit hyper-sphere, which can have several directions that do not have good discriminative power. Finally, sampling projecting directions in high-dimension is also time-consuming and memory-consuming. As a consequence, avoiding the vectorization step can improve the efficiency of the whole process.

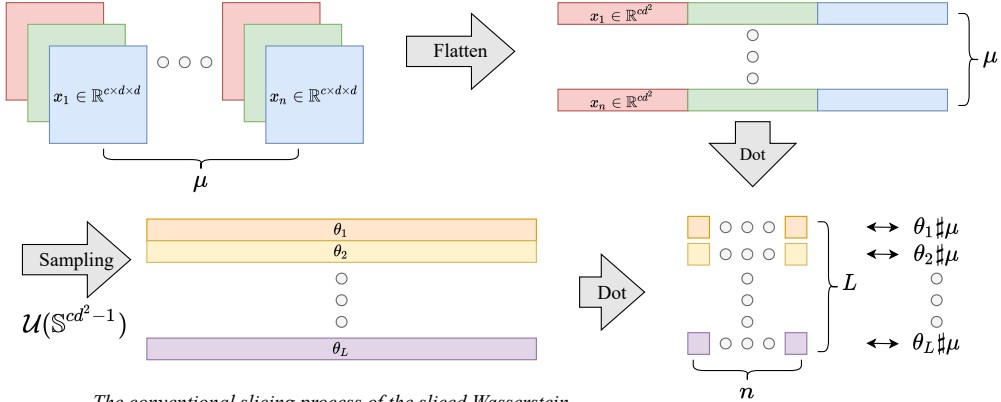

*The conventional slicing process of the sliced Wasserstein*

Figure 1: The conventional slicing process of sliced Wasserstein distance. The images $X_1, \ldots, X_n \in \mathbb{R}^{c \times d \times d}$ are first flattened into vectors in $\mathbb{R}^{cd^2}$ and then the Radon transform is applied to these vectors to lead to sliced Wasserstein (1) on images.

**Convolution operator:** We now define the convolution operator on tensors [16], which will be used as an alternative way of projecting images to one dimension in the sliced Wasserstein. The definition of the convolution operator with stride and dilation is as follows.

**Definition 1** *(Convolution) Given the number of channels $c \geq 1$, the dimension $d \geq 1$, the stride size $s \geq 1$, the dilation size $b \geq 1$, the size of kernel $k \geq 1$, the convolution of a tensor $X \in \mathbb{R}^{c \times d \times d}$ with a kernel size $K \in \mathbb{R}^{c \times k \times k}$ is $X \overset{s,b}{*} K = Y$, $Y \in \mathbb{R}^{1 \times d' \times d'}$ where $d' = \frac{d - b(k-1) - 1}{s} + 1$. For $i = 1, \ldots, d'$ and $j = 1, \ldots, d'$, $Y_{1,i,j}$ is defined as:*
$$Y_{1,i,j} = \sum_{h=1}^{c} \sum_{i'=0}^{k-1} \sum_{j'=0}^{k-1} X_{h,s(i-1)+bi'+1,s(j-1)+bj'+1} \cdot K_{h,i'+1,j'+1}.$$

From its definition, we can check that the computational complexity of the convolution operator is $\mathcal{O}\left( c \left( \frac{d - b(k-1) - 1}{s} + 1 \right)^2 k^2 \right)$.

## 3 Convolution Sliced Wasserstein

In this section, we will define a convolution slicer that maps a tensor to a scalar by convolution operators. Moreover, we discuss the convolution slicer and some of its specific forms including the convolution-base slicer, the convolution-stride slicer, the convolution-dilation slicer, and their non-linear extensions. After that, we derive the convolution sliced Wasserstein (CSW), a family of variants of sliced Wasserstein, that utilizes a convolution slicer as the projecting method. Finally, we discuss some theoretical properties of CSW, namely, its metricity, its computational complexity, its sample complexity, and its connection to other variants of sliced Wasserstein.

### 3.1 Convolution Slicer

We first start with the definition of the convolution slicer, which plays an important role in defining convolution sliced Wasserstein.

**Definition 2** *(Convolution Slicer) For $N \geq 1$, given a sequence of kernels $K^{(1)} \in \mathbb{R}^{c^{(1)} \times d^{(1)} \times d^{(1)}}, \ldots, K^{(N)} \in \mathbb{R}^{c^{(N)} \times d^{(N)} \times d^{(N)}}$, a convolution slicer $\mathcal{S}(\cdot | K^{(1)}, \ldots, K^{(N)})$ on $\mathbb{R}^{c \times d \times d}$ is a composition of $N$ convolution functions with kernels $K^{(1)}, \ldots, K^{(N)}$ (with stride or dilation if needed) such that $\mathcal{S}(X | K^{(1)}, \ldots, K^{(N)}) \in \mathbb{R} \quad \forall X \in \mathbb{R}^{c \times d \times d}$.*

As indicated in Definition 2, the idea of the convolution slicer is to progressively map a given data $X$ to a one-dimensional subspace through a sequence of convolution kernels, which capture spatial relations across channels as well as local information of the data. It is starkly different from the vectorization step in standard sliced Wasserstein on images (1). The illustration of the convolution slicer is given in Figure 2.

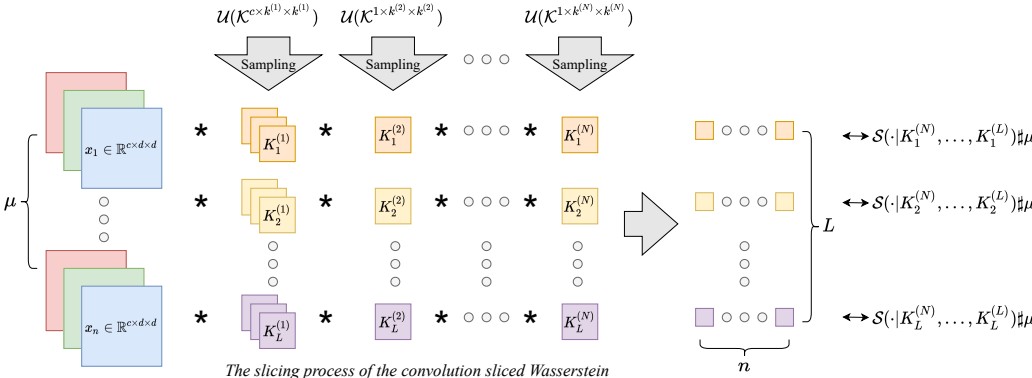

*The slicing process of the convolution sliced Wasserstein*

Figure 2: The convolution slicing process (using the convolution slicer). The images $X_1, \ldots, X_n \in \mathbb{R}^{c \times d \times d}$ are directly mapped to a scalar by a sequence of convolution functions which have kernels as random tensors. This slicing process leads to the convolution sliced Wasserstein on images.

We consider three particular types of convolution slicers based on using linear function on the convolution operator, named convolution-base, convolution-stride, and convolution-dilation slicers. We defer the definition of convolution-dilation slicers to Definition 5. We first start with the definition of the convolution-base slicer.

**Definition 3** *(Convolution-base Slicer) Given* $X \in \mathbb{R}^{c \times d \times d}$ *(d $\geq$ 2),*

*1. When $d$ is even, $N$ is the biggest integer that satisfies $d = 2^{N-1} a$ with $a$ is also an integer, sliced kernels are defined as $K^{(1)} \in \mathbb{R}^{c \times (2^{-1}d+1) \times (2^{-1}d+1)}$ and $K^{(h)} \in \mathbb{R}^{1 \times (2^{-h}d+1) \times (2^{-h}d+1)}$ for $h = 2, \ldots, N-1$, and $K^{(N)} \in \mathbb{R}^{1 \times a \times a}$ where $a = \frac{d}{2^{N-1}}$. Then, the* convolution-base slicer $\mathcal{CS}\text{-}b(X|K^{(1)}, \ldots, K^{(N)})$ *is defined as:*

$$\mathcal{CS}\text{-}b(X|K^{(1)}, \ldots, K^{(N)}) = X^{(N)}, \quad X^{(h)} = \begin{cases} X & h = 0 \\ X^{(h-1)} \overset{1,1}{*} K^{(h)} & 1 \leq h \leq N, \end{cases}$$

*2. When $d$ is odd, the* convolution-base slicer $\mathcal{CS}\text{-}b(X|K^{(1)}, \ldots, K^{(N)})$ *takes the form:*

$$\mathcal{CS}\text{-}b(X|K^{(1)}, \ldots, K^{(N)}) = \mathcal{CS}\text{-}b(X \overset{1,1}{*} K^{(1)} | K^{(2)}, \ldots, K^{(N)}),$$

*where $K^{(1)} \in \mathbb{R}^{c \times 2 \times 2}$ and $K^{(2)}, \ldots, K^{(N)}$ are the corresponding sliced kernels that are defined on the dimension $d - 1$.*

The idea of the convolution-base slicer in Definition 3 is to reduce the width and the height of the image by half after each convolution operator. If the width and the height of the image are odd, the first convolution operator is to reduce the size of the image by one via convolution with kernels of size $2 \times 2$, and then the same procedure as that of the even case is applied. We would like to remark that the conventional slicing of sliced Wasserstein in Section 2 is equivalent to a convolution-base slicer $\mathcal{S}(\cdot|K^{(1)})$ where $K^{(1)} \in \mathbb{R}^{c \times d \times d}$ that satisfies the constraint $\sum_{h=1}^{c} \sum_{i=1}^{d} \sum_{j=1}^{d} K_{h,i,j}^{(1)2} = 1$.

We now discuss the second variant of the convolution slicer, named convolution-stride slicer, where we further incorporate stride into the convolution operators. Its definition is as follows.

**Definition 4** *(Convolution-stride Slicer) Given* $X \in \mathbb{R}^{c \times d \times d}$ *(d $\geq$ 2),*

*1. When $d$ is even, $N$ is the biggest integer that satisfies $d = 2^{N-1} a$ with $a$ is also an integer, sliced kernels are defined as $K^{(1)} \in \mathbb{R}^{c \times 2 \times 2}$ and $K^{(h)} \in \mathbb{R}^{1 \times 2 \times 2}$ for $h = 2, \ldots, N-1$, and $K^{(N)} \in \mathbb{R}^{1 \times a \times a}$ where $a = \frac{d}{2^{N-1}}$. Then, the* convolution-stride slicer $\mathcal{CS}\text{-}s(X|K^{(1)}, \ldots, K^{(N)})$ *is defined as:*

$$\mathcal{CS}\text{-}s(X|K^{(1)}, \ldots, K^{(N)}) = X^{(N)}, \quad X^{(h)} = \begin{cases} X & h = 0 \\ X^{(h-1)} \overset{2,1}{*} K^{(h)} & 1 \leq h \leq N-1, \\ X^{(h-1)} \overset{1,1}{*} K^{(h)} & h = N, \end{cases}$$

*2. When $d$ is odd, the* convolution-stride slicer $\mathcal{CS}\text{-}s(X|K^{(1)}, \ldots, K^{(N)})$ *takes the form:*

$$\mathcal{CS}\text{-}s(X|K^{(1)}, \ldots, K^{(N)}) = \mathcal{CS}\text{-}s(X \overset{1,1}{*} K^{(1)}|K^{(2)}, \ldots, K^{(N)}),$$

*where $K^{(1)} \in \mathbb{R}^{c \times 2 \times 2}$ and $K^{(2)}, \ldots, K^{(N)}$ are the corresponding sliced kernels that are defined on the dimension $d - 1$.*

Similar to the convolution-base slicer in Definition 3, the convolution-stride slicer reduces the width and the height of the image by half after each convolution operator. We use the same procedure of reducing the height and the width of the image by one when the height and the width of the image are odd. The benefit of the convolution-stride slicer is that the size of its kernels does not depend on the width and the height of images as that of the convolution-base slicer. This difference improves the computational complexity and time complexity of the convolution-stride slicer over those of the convolution-base slicer (cf. Proposition 1).

**Definition 5** *(Convolution-dilation Slicer) Given $X \in \mathbb{R}^{c \times d \times d}$ ($d \geq 2$),*

1. *When $d$ is even, $N$ is the biggest integer that satisfies $d = 2^{N-1}a$ with $a$ is also an integer, sliced kernels are defined as $K^{(1)} \in \mathbb{R}^{c \times 2 \times 2}$ and $K^{(h)} \in \mathbb{R}^{1 \times 2 \times 2}$ for $h = 2, \ldots, N-1$, and $K^{(N)} \in \mathbb{R}^{1 \times a \times a}$ where $a = \frac{d}{2^{N-1}}$. Then, the* convolution-dilation slicer $\mathcal{CS}\text{-}d(X|K^{(1)}, \ldots, K^{(N)})$ *is defined as:*

$$\mathcal{CS}\text{-}d(X|K^{(1)}, \ldots, K^{(N)}) = X^{(N)}, \quad X^{(h)} = \begin{cases} X & h = 0 \\ X^{(h-1)} \overset{1,d/2^h}{*} K^{(h)} & 1 \leq h \leq N-1, \\ X^{(h-1)} \overset{1,1}{*} K^{(h)} & h = N, \end{cases}$$

2. *When $d$ is odd, the* convolution-dilation slicer $\mathcal{CS}\text{-}d(X|K^{(1)}, \ldots, K^{(N)})$ *takes the form:*

$$\mathcal{CS}\text{-}d(X|K^{(1)}, \ldots, K^{(N)}) = \mathcal{CS}\text{-}d(X \overset{1,1}{*} K^{(1)}|K^{(2)}, \ldots, K^{(N)}),$$

*where $K^{(1)} \in \mathbb{R}^{c \times 2 \times 2}$ and $K^{(2)}, \ldots, K^{(N)}$ are the corresponding sliced kernels that are defined on the dimension $d - 1$.*

As with the previous slicers, the convolution-dilation slicer also reduces the width and the height of the image by half after each convolution operator and it uses the same procedure for the odd dimension cases. The design of kernels' size of the convolution-dilation slicer is the same as that of the convolution-stride slicer. However, the convolution-dilation slicer has a bigger receptive field in each convolution operator which might be appealing when the information of the image is presented by a big block of pixels.

**Computational and projection memories complexities of the convolution slicers:** We now establish the computational and projection memory complexities of convolution-base, convolution-stride, and convolution-dilation slicers in the following proposition. We would like to recall that the projection memory complexity is the memory that is needed to store a slice (convolution kernels).

**Proposition 1** *(a) When $d$ is even, $N$ is the biggest integer that satisfies $d = 2^{N-1}a$ with $a$ is also an integer, and $N = [\log_2 d]$, the computational and projection memory complexities of convolution-base slicer are respectively at the order of $\mathcal{O}(cd^4)$ and $\mathcal{O}(cd^2)$. When $d$ is odd, these complexities are at the order of $\mathcal{O}(cd^2 + d^4)$ and $\mathcal{O}(c + d^2)$.*

*(b) The computational and projection memory complexities of convolution-stride slicer are respectively at the order of $\mathcal{O}(cd^2)$ and $\mathcal{O}(c + [\log_2 d])$.*

*(c) The computational and projection memory complexities of convolution-dilation slicer are respectively at the order of $\mathcal{O}(cd^2)$ and $\mathcal{O}(c + [\log_2 d])$.*

Proof of Proposition 1 is in Appendix B.4. We recall that the computational complexity and the projection memory complexity of the conventional slicing in sliced Wasserstein are $\mathcal{O}(cd^2)$ and $\mathcal{O}(cd^2)$. We can observe that the convolution-base slicer has a worse computational complexity than the conventional slicing while having the same projection memory complexity. Since the size of

kernels does not depend on the size of images, the convolution-stride slicer and the convolution-dilation slicer have the same computational complexity as the conventional slicing $\mathcal{O}(cd^2)$. However, their projection memory complexities are cheaper than conventional slicing, namely, $\mathcal{O}(c + [\log_2 d])$ compared to $\mathcal{O}(cd^2)$.

**Non-linear convolution-base slicer:** The composition of convolution functions in the linear convolution slicer and its linear variants is still a linear function, which may not be effective when the data lie in a complex and highly non-linear low-dimensional subspace. A natural generalization of linear convolution slicers to enhance the ability of the slicers to capture the non-linearity of the data is to apply a non-linear activation function after convolution operators. This enables us to define a non-linear slicer in Definition 7 in Appendix C. The non-linear slicer can be seen as a defining function in generalized Radon Transform [52] which was used in generalized sliced Wasserstein [24].

## 3.2 Convolution Sliced Wasserstein

Given the definition of convolution slicers, we now state general definition of convolution sliced Wasserstein. An illustration of the convolution sliced Wasserstein is given in Figure 2.

**Definition 6** *For any $p \geq 1$, the* convolution sliced Wasserstein *(CSW) of order $p > 0$ between two given probability measures $\mu, \nu \in \mathcal{P}_p(\mathbb{R}^{c \times d \times d})$ is given by:*

$$CSW_p(\mu, \nu) := \left( \mathbb{E} \left[ W_p^p \left( \mathcal{S}(\cdot | K^{(1)}, \ldots, K^{(N)}) \sharp \mu, \mathcal{S}(\cdot | K^{(1)}, \ldots, K^{(N)}) \sharp \nu \right) \right] \right)^{\frac{1}{p}},$$

*where the expectation is taken with respect to $K^{(1)} \sim \mathcal{U}(\mathcal{K}^{(1)}), \ldots, K^{(N)} \sim \mathcal{U}(\mathcal{K}^{(N)})$. Here, $\mathcal{S}(\cdot | K^{(1)}, \ldots, K^{(N)})$ is a convolution slicer with $K^{(l)} \in \mathbb{R}^{c^{(l)} \times k^{(l)} \times k^{(l)}}$ for any $l \in [N]$ and $\mathcal{U}(\mathcal{K}^{(l)})$ is the uniform distribution with the realizations being in the set $\mathcal{K}^{(l)}$ which is defined as $\mathcal{K}^{(l)} := \left\{ K^{(l)} \in \mathbb{R}^{c^{(l)} \times k^{(l)} \times k^{(l)}} | \sum_{h=1}^{c^{(l)}} \sum_{i'=1}^{k^{(l)}} \sum_{j'=1}^{k^{(l)}} K_{h,i',j'}^{(i)2} = 1 \right\}$, namely, the set $\mathcal{K}^{(l)}$ consists of tensors $K^{(l)}$ whose squared $\ell_2$ norm is 1.*

The constraint that $\ell_2$ norms of $K^{(l)}$ is 1 is for guaranteeing the distances between projected supports are bounded. When we specifically consider the convolution slicer as convolution-base slicer ($\mathcal{CS}$-b), convolution-stride slicer ($\mathcal{CS}$-s), and convolution-dilation slicer ($\mathcal{CS}$-d), we have the corresponding notions of convolution-base sliced Wasserstein (CSW-b), convolution-stride sliced Wasserstein (CSW-s), and convolution-dilation sliced Wasserstein (CSW-d).

**Monte Carlo estimation and implementation:** Similar to the conventional sliced Wasserstein, the expectation with respect to kernels $K^{(1)}, \ldots, K^{(N)}$ uniformly drawn from the sets $\mathcal{K}^{(1)}, \ldots, \mathcal{K}^{(N)}$ in the convolution sliced Wasserstein is intractable to compute. Therefore, we also make use of Monte Carlo method to approximate the expectation, which leads to the following approximation of the convolution sliced Wasserstein:

$$\text{CSW}_p^p(\mu, \nu) \approx \frac{1}{L} \sum_{i=1}^{L} W_p^p \left( \mathcal{S}(\cdot | K_i^{(1)}, \ldots, K_i^{(N)}) \sharp \mu, \mathcal{S}(\cdot | K_i^{(1)}, \ldots, K_i^{(N)}) \sharp \nu \right), \qquad (2)$$

where $K_i^{(\ell)}$ are uniform samples from the sets $\mathcal{K}^{(\ell)}$ (which is equivalent to sample uniformly from $\mathbb{S}^{c^{(l)} \cdot k^{(l)2}}$ then applying the one-to-one reshape mapping) for any $\ell \in [N]$ and $i \in [L]$. Since each of the convolution slicer $\mathcal{S}(\cdot | K_i^{(1)}, \ldots, K_i^{(N)})$ is in one dimension, we can utilize the closed-form expression of Wasserstein metric in one dimension to compute $W_p \left( \mathcal{S}(\cdot | K_i^{(1)}, \ldots, K_i^{(N)}) \sharp \mu, \mathcal{S}(\cdot | K_i^{(1)}, \ldots, K_i^{(N)}) \sharp \nu \right)$ with a complexity of $\mathcal{O}(m \log_2 m)$ for each $i \in [L]$ where $m$ is the maximum number of supports of $\mu$ and $\nu$. Therefore, the total computational complexity of computing the Monte Carlo approximation (2) is $\mathcal{O}(Lm \log_2 m)$ when the probability measures $\mu$ and $\nu$ have at most $m$ supports. It is comparable to the computational complexity of sliced Wasserstein on images (1) where we directly vectorize the images and apply the Radon transform to these flatten images. Finally, for the implementation, we would like to remark that $L$ convolution slicers in equation (2) can be computed *independently* and *parallelly* using the group convolution implementation which is supported in almost all libraries.

**Properties of convolution sliced Wasserstein:** We first have the following result for the metricity of the convolution sliced Wasserstein.

**Theorem 1** *For any $p \geq 1$, the convolution sliced Wasserstein $CSW_p(.,.)$ is a pseudo-metric on the space of probability measures on $\mathbb{R}^{c \times d \times d}$, namely, it is symmetric, and satisfies the triangle inequality.*

Proof of Theorem 1 is in Appendix B.1. We would like to mention that CSW can might still be a metric since the convolution slicer might be injective. Our next result establishes the connection between the convolution sliced Wasserstein and max-sliced Wasserstein and Wasserstein distances.

**Proposition 2** *For any $p \geq 1$, we find that $CSW_p(\mu, \nu) \leq \text{Max-SW}_p(\mu, \nu) \leq W_p(\mu, \nu)$, where $\text{Max-SW}_p(\mu, \nu) := \max_{\theta \in \mathbb{R}^{cd^2}: \|\theta\| \leq 1} W_p(\theta \sharp \mu, \theta \sharp \nu)$ is max-sliced Wasserstein of order $p$.*

Proof of Proposition 2 is in Appendix B.2. Given the bounds in Proposition 2, we demonstrate that the convolution sliced Wasserstein does not suffer from the curse of dimensionality for the inference purpose, namely, the sample complexity for the empirical distribution from i.i.d. samples to approximate their underlying distribution is at the order of $\mathcal{O}(n^{-1/2})$.

**Proposition 3** *Assume that $P$ is a probability measure supported on compact set of $\mathbb{R}^{c \times d \times d}$. Let $X_1, X_2, \ldots, X_n$ be i.i.d. samples from $P$ and we denote $P_n = \frac{1}{n} \sum_{i=1}^{n} \delta_{X_i}$ as the empirical measure of these data. Then, for any $p \geq 1$, there exists a universal constant $C > 0$ such that*

$$\mathbb{E}[CSW_p(P_n, P)] \leq C\sqrt{(cd^2 + 1)\log n / n},$$

*where the outer expectation is taken with respect to the data $X_1, X_2, \ldots, X_n$.*

Proof of Proposition 3 is in Appendix B.3. The result of Proposition 3 indicates that the sample complexity of the convolution sliced Wasserstein is comparable to that of the sliced Wasserstein on images (1), which is at the order of $\mathcal{O}(n^{-1/2})$ [4], and better than that of the Wasserstein metric, which is at the order of $\mathcal{O}(n^{-1/(2cd^2)})$ [15].

**Extension to non-linear convolution sliced Wasserstein:** In Appendix C, we provide a non-linear version of the convolution sliced Wasserstein, named non-linear convolution sliced Wasserstein. The high-level idea of the non-linear version is to incorporate non-linear activation functions to the convolution-base, convolution-stride, and convolution-dilation slicers. The inclusion of non-linear activation functions is to enhance the ability of slicers to capture the non-linearity of the data. By plugging these non-linear convolution slicers into the general definition of the convolution sliced Wasserstein in Definition 6, we obtain the non-linear variants of convolution sliced Wasserstein.

## 4 Experiments

In this section, we focus on comparing the sliced Wasserstein (SW) (with the conventional slicing), the convolution-base sliced Wasserstein (CSW-b), the convolution sliced Wasserstein with stride (CSW-s), and the convolution sliced Wassersstein with dilation (CSW-d) in training generative models on standard benchmark image datasets such as CIFAR10 (32x32) [27], STL10 (96x96) [8], CelebA (64x64), and CelebA-HQ (128x128) [37]. We recall that the number of projections in SW and CSW's variants is denoted as $L$. Finally, we also show the values of the SW and the CSW variants between probability measures over digits of the MNIST dataset [30] in Appendix D.1. From experiments on MNIST, we observe that values of CSW variants are similar to values of SW while having better projection complexities.

In generative modeling, we follow the framework of the sliced Wasserstein generator in [13] with some modifications of neural network architectures. The details of the training are given in Appendix D.2. We train the above model on standard benchmarks such as CIFAR10 (32x32) [27], STL10 (96x96) [8], CelebA (64x64), and CelebAHQ (128x128) [37]. To compare models, we use the FID score [20] and the Inception score (IS) [55]. The detailed settings about architectures, hyperparameters, and evaluation of FID and IS are given in Appendix E. We first show the FID scores and IS scores of generative models trained by SW and CSW's variants with the number of projections $L \in \{1, 100, 1000\}$ in Table 1. In the table, we report the performance of models at the last training epoch. We do not report the IS scores on CelebA and CelebA-HQ since the IS scores are not suitable for face images. We then demonstrate the FID scores and IS scores across training epochs in Figure 3 for investigating the convergence of generative models trained by SW and CSW's variants. After

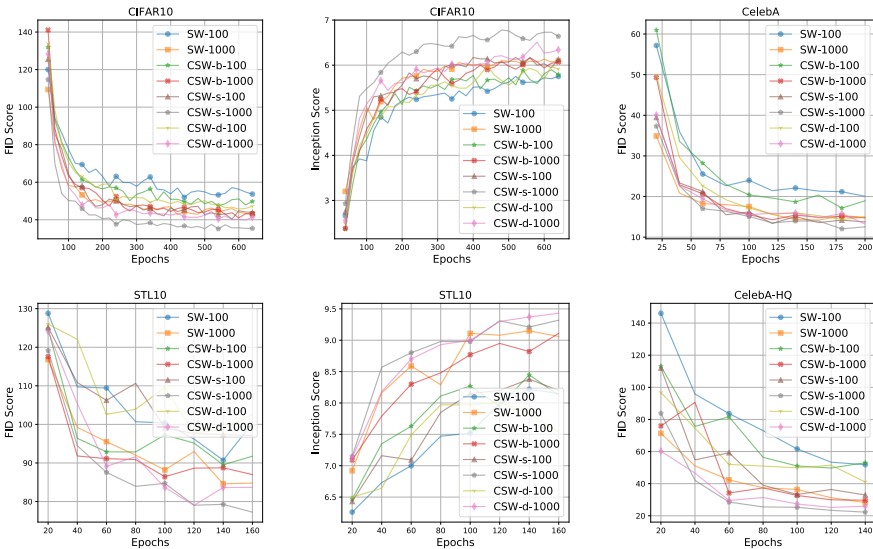

Figure 3: FID scores and IS scores over epochs of different training losses on datasets. We observe that CSW's variants usually help the generative models converge faster.

Table 1: Summary of FID and IS scores of methods on CIFAR10 (32x32), CelebA (64x64), STL10 (96x96), and CelebA-HQ (128x128). Some results on CIFAR10 are reported from 5 different runs.

| Method | CIFAR10 (32x32) | | CelebA (64x64) | STL10 (96x96) | | CelebA-HQ (128x128) |
|---|---|---|---|---|---|---|
| | FID ($\downarrow$) | IS ($\uparrow$) | FID ($\downarrow$) | FID ($\downarrow$) | IS ($\uparrow$) | FID ($\downarrow$) |
| SW (L=1) | 87.97 | 3.59 | 128.81 | 170.96 | 3.68 | **275.44** |
| CSW-b (L=1) | 84.38 | 4.28 | 85.83 | 173.33 | **3.89** | 315.91 |
| CSW-s (L=1) | 80.10 | 4.31 | **66.52** | **168.9**3 | 3.75 | 303.57 |
| CSW-d (L=1) | **63.94** | **4.89** | 89.37 | 212.61 | 2.48 | 321.06 |
| SW (L=100) | 52.36±0.76 | 5.79±0.16 | 20.08 | 100.35 | 8.14 | 51.80 |
| CSW-b (L=100) | 49.67±2.00 | 5.87±0.15 | 18.96 | **91.75** | 8.11 | 53.05 |
| CSW-s (L=100) | **43.73±2.09** | **6.17±0.06** | **13.76** | 97.08 | **8.20** | **32.94** |
| CSW-d (L=100) | 47.23±1.12 | 5.97±0.11 | 14.96 | 102.58 | 7.53 | 41.01 |
| SW (L=1000) | 44.25±1.21 | 6.02±0.03 | 14.92 | 84.78 | 9.06 | 28.19 |
| CSW-b (L=1000) | 42.88±0.98 | 6.11±0.10 | 14.75 | 86.98 | 9.11 | 29.69 |
| CSW-s (L=1000) | **36.80±1.44** | **6.55±0.12** | **12.55** | **77.24** | 9.31 | **22.25** |
| CSW-d (L=1000) | 40.44±1.02 | 6.38±0.14 | 13.24 | 83.36 | **9.42** | 25.93 |

that, we report the training time and training memory of SW and CSW variants in Table 5. Finally, we show randomly generated images from SW's models and CSW-s' models on CelebA dataset in Figure 4. Generated images of all models on all datasets are given in Figures 5-8 in Appendix D.2.

**Summary of FID scores and IS scores:** According to Table 1, on CIFAR10, CSW-d gives the lowest values of FID scores and IS scores when $L = 1$ while CSW-s gives the lowest FID scores when $L = 100$ and $L = 1000$. Compared to CSW-s, CSW-d and CSW-b yield higher FID scores and lower IS scores. However, CSW-d and CSW-b are still better than SW. On CelebA, CSW-s performs the best in all settings. On STL10, CSW's variants are also better than the vanilla SW; however, it is unclear which is the best variant. On CelebA-HQ, SW gives the lowest FID score when $L = 1$. In contrast, when $L = 100$ and $L = 1000$, CSW-s is the best choice for training the generative model. Since the FID scores of $L = 1$ are very high on CelebA-HQ and STL10, the scores are not very meaningful for comparing SW and CSW's variants. For all models, increasing $L$ leads to better quality. Overall, we observe that CSW's variants enhance the performance of generative models.

**FID scores and IS scores across epochs:** From Figure 3, we observe that CSW's variants help the generative models converge faster than SW when $L = 100$ and $L = 1000$. Increasing the number of projections from 100 to 1000, the generative models from both SW and CSW's variants become better. Overall, CSW-s is the best option for training generative models among CSW's variants since its FID curves are the lowest and its IS curves are the highest.

**Generated images:** We show randomly generated images on CelebA dataset in Figure 4 and Figure 6 (Appendix D), and generated images on CIFAR10, CelebA, STL10, and CelebA-HQ in Figures 5-8 as

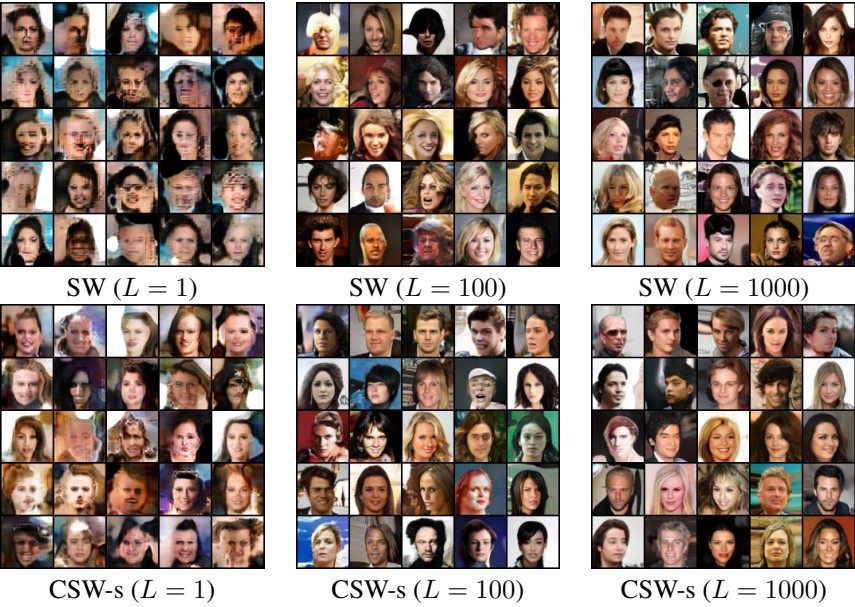

| SW ($L=1$) | SW ($L=100$) | SW ($L=1000$) |
| CSW-s ($L=1$) | CSW-s ($L=100$) | CSW-s ($L=1000$) |

Figure 4: Random generated images of SW and CSW-s on CelebA.

qualitative comparison between SW and CSW variants. From the figures, we can see that generated images of CSW-s is more realistic than ones of SW. The difference is visually clear when the number of projections $L$ is small e.g., $L=1$ and $L=100$. When $L=1000$, we can still figure out that CSW-s is better than SW by looking at the sharpness of the generated images. Also, we can visually observe the improvement of SW and CSW-s when increasing the number of projections. In summary, the qualitative results are consistent with the quantitative results (FID scores and IS scores) in Table 1. For the generated images of CSW-b and CSW-d, we also observe the improvement compared to the SW which is consistent with the improvement of FID scores and IS scores.

**Non-linear convolution sliced Wasserstein:** We also compare non-linear extensions of SW and CSW variants in training generative models on CIFAR10 in Appendix D. For details of non-linear extensions, we refer to Appendix C. From experiments, we observe that convolution can also improve the performance of sliced Wasserstein in non-linear projecting cases. Compared to linear versions, non-linear versions can enhance the quality of the generative model or yield comparable results.

## 5 Conclusion

We have addressed the issue of the conventional slicing process of sliced Wasserstein when working with probability measures over images. In particular, sliced Wasserstein is defined on probability measures over vectors which leads to the step of vectorization for images. As a result, the conventional slicing process cannot exploit the spatial structure of data for designing the space of projecting directions and projecting operators. To address the issue, we propose a new slicing process by using the convolution operator which has been shown to be efficient on images. Moreover, we investigate the computational complexity and projection memory complexity of the new slicing technique. We show that convolution slicing is comparable to conventional slicing in terms of computational complexity while being better in terms of projection memory complexity. By utilizing the new slicing technique, we derive a novel family of sliced Wassersein variants, named convolution sliced Wasserstein. We investigate the properties of the convolution sliced Wasserstein including its metricity, its computational and sample complexities, and its connection to other variants of sliced Wasserstein in literature. Finally, we carry out extensive experiments in comparing digits images and training generative models on standard benchmark datasets to demonstrate the favorable performance of the convolution sliced Wasserstein.

## Acknowledgements

NH acknowledges support from the NSF IFML 2019844 and the NSF AI Institute for Foundations of Machine Learning.

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
