# Supplement to "Revisiting Sliced Wasserstein on Images: From Vectorization to Convolution"

In this supplement, we first discuss related works and the potential impacts and limitations of our works in Appendix A. In Appendix B, we provide proofs for key results in the paper. In Appendix C, we introduce non-linear versions of the convolution sliced Wasserstein, max convolution sliced Wasserstein, and convolution projected robust Wasserstein. In Appendix D, we include additional experiments for comparing measures over MNIST's digits via sliced Wasserstein and convolution sliced Wasserstein. Also, we further provide generated images for convolution sliced Wasserstein under generative model settings , and generative experiemnts on max convolution sliced Wasserstein and convolution projected robust Wasserstein. Finally, in Appendix E, we include details of experimental settings in the paper.

## A    Related Works, Potential Impact, and Limitations

Sliced Wasserstein is used for the pooling mechanism for aggregating a set of features in [40]. Sliced Wasserstein gradient flows are investigated in [38, 5]. Variational inference based on sliced Wasserstein is carried out in [64]. Similarly, sliced Wasserstein is used for approximate Bayesian computation in [41]. Statistical guarantees of training generative models with sliced Wasserstein is derived in [43]. Other frameworks for generative modeling using sliced Wasserstein are sliced iterative normalizing flows [11] and run-sort-rerun for fine-tuning pre-trained model [32]. Differentially private sliced Wasserstein is proposed in [53]. Approximating Wasserstein distance based on one-dimensional transportation plans from orthogonal projecting directions is introduced in [54]. To reduce the projection complexity of sliced Wasserstein, a biased approximation based on the concentration of Gaussian projections is proposed in [42]. Augmenting probability measures to a higher-dimensional space for a better linear separation is used in augmented sliced Wasserstein [7]. Projected Robust Wasserstein (PRW) metrics that find the best orthogonal linear projecting operator onto $k > 1$ dimensional space and Riemannian optimization techniques for solving it are proposed in [48, 33, 22]. Sliced Gromov Wasserstein, a fast sliced version of Gromov Wasserstein, is proposed in [57]. The slicing technique is also be applied in approximating mutual information [17]. We would like to recall that all the above works assume working with vector spaces and need to use vectorization when dealing with images. In [56], convolution is used for learning the ground cost metric of optimal transport while it is used to project measures to one-dimensional measures in our work.

**Potential Impact:** This work addresses a fundamental problem of designing a slicing process for sliced Wasserstein on images and it can be used in various applications that perform on images. Therefore, it could create negative potential impacts if it is used in applications that do not have good purposes.

**Limitations:** One limitation of CSW is that it is a pseudo metric on the space of all distribution over tensors. However, this is because we do not assume any structure on distribution over images. In practice, many empirical investigations show that image datasets belong to some geometry group (symmetry, rotation invariant, translation invariant, and so on). Therefore, the set of distributions over images might be a subset of the set of distributions over tensors. If the convolutional transform can hold the injectivity on the set of distributions over images, CSW can be a metric on the space of distributions over images. In our applications, we compare the value of sliced Wasserstein and convolution sliced Wasserstein on MNIST digits in Table 4 in Appendix D.1, we found that the values of CSW are closed to the value of SW that can be considered as a test for our hypothesis of metricity of CSW. To our best knowledge, there is no formal definition of the space of distributions over images and its property. Therefore, we will leave this for future work.

In deep learning applications, sliced Wasserstein is computed between empirical distributions over mini-batches of samples that are randomly drawn from the original distribution [1]. This is known as mini-batch optimal transport with sliced Wasserstein kernel that is used when dealing with very large scale distributions and implicit continuous distributions. When using mini-batches, both Wasserstein distance, sliced Wasserstein distance, and convolutional sliced Wasserstein will lose its metricity to become a loss [14]. Therefore, metricity is not the deciding factor in some applications of sliced Wasserstein such as deep generative model, deep domain adaptation, and so on. This partially explains the better performance of CSW on our deep generative model experiments in Table 1.

# B Proofs

In this appendix, we provide proofs for key results in the main text.

## B.1 Proof of Theorem 1

For any $p \geq 1$, it is clear that when $\mu = \nu$, then $\mathrm{CSW}_p(\mu, \nu) = 0$. Furthermore, $\mathrm{CSW}_p(\mu, \nu) = \mathrm{CSW}_p(\nu, \mu)$ for any probability measures $\mu$ and $\nu$. Therefore, to obtain the conclusion of the theorem, it is sufficient to demonstrate that is satisfies the triangle inequality. Indeed, for any probability measures $\mu_1, \mu_2, \mu_3$, we find that

$$\mathrm{CSW}_p(\mu_1, \mu_3)$$
$$= \left( \mathbb{E}_{K^{(1)} \sim \mathcal{U}(\mathcal{K}^{(1)}), \ldots, K^{(N)} \sim \mathcal{U}(\mathcal{K}^{(N)})} \left[ W_p^p \left( \mathcal{S}(\cdot | K^{(1)}, \ldots, K^{(N)}) \sharp \mu_1, \mathcal{S}(\cdot | K^{(1)}, \ldots, K^{(N)}) \sharp \mu_3 \right) \right] \right)^{\frac{1}{p}}$$
$$\leq \left( \mathbb{E}_{K^{(1)} \sim \mathcal{U}(\mathcal{K}^{(1)}), \ldots, K^{(N)} \sim \mathcal{U}(\mathcal{K}^{(N)})} \left[ W_p \left( \mathcal{S}(\cdot | K^{(1)}, \ldots, K^{(N)}) \sharp \mu_1, \mathcal{S}(\cdot | K^{(1)}, \ldots, K^{(N)}) \sharp \mu_2 \right) \right. \right.$$
$$\left. \left. + W_p \left( \mathcal{S}(\cdot | K^{(1)}, \ldots, K^{(N)}) \sharp \mu_2, \mathcal{S}(\cdot | K^{(1)}, \ldots, K^{(N)}) \sharp \mu_3 \right) \right]^p \right)^{1/p}$$
$$\leq \left( \mathbb{E}_{K^{(1)} \sim \mathcal{U}(\mathcal{K}^{(1)}), \ldots, K^{(N)} \sim \mathcal{U}(\mathcal{K}^{(N)})} \left[ W_p^p \left( \mathcal{S}(\cdot | K^{(1)}, \ldots, K^{(N)}) \sharp \mu_1, \mathcal{S}(\cdot | K^{(1)}, \ldots, K^{(N)}) \sharp \mu_2 \right) \right] \right)^{1/p}$$
$$+ \left( \mathbb{E}_{K^{(1)} \sim \mathcal{U}(\mathcal{K}^{(1)}), \ldots, K^{(N)} \sim \mathcal{U}(\mathcal{K}^{(N)})} \left[ W_p^p \left( \mathcal{S}(\cdot | K^{(1)}, \ldots, K^{(N)}) \sharp \mu_2, \mathcal{S}(\cdot | K^{(1)}, \ldots, K^{(N)}) \sharp \mu_3 \right) \right] \right)^{1/p}$$
$$= \mathrm{CSW}_p(\mu_1, \mu_2) + \mathrm{CSW}_p(\mu_2, \mu_3),$$

where the first inequality is due to the triangle inequality with Wasserstein metric, namely, we have

$$W_p \left( \mathcal{S}(\cdot | K^{(1)}, \ldots, K^{(N)}) \sharp \mu_1, \mathcal{S}(\cdot | K^{(1)}, \ldots, K^{(N)}) \sharp \mu_3 \right)$$
$$\leq W_p \left( \mathcal{S}(\cdot | K^{(1)}, \ldots, K^{(N)}) \sharp \mu_1, \mathcal{S}(\cdot | K^{(1)}, \ldots, K^{(N)}) \sharp \mu_2 \right)$$
$$+ W_p \left( \mathcal{S}(\cdot | K^{(1)}, \ldots, K^{(N)}) \sharp \mu_2, \mathcal{S}(\cdot | K^{(1)}, \ldots, K^{(N)}) \sharp \mu_3 \right)$$

while the second inequality is an application of Minkowski inequality for integrals. As a consequence, we obtain the conclusion of the theorem.

## B.2 Proof of Proposition 2

The proof of this proposition is direct from the definition of the convolution sliced Wasserstein. Here, we provide the proof for the completeness. Indeed, since the convolution slicer $\mathcal{S}(\cdot | K^{(1)}, \ldots, K^{(N)})$ is a mapping from $\mathbb{R}^{c \times d \times d}$ to $\mathbb{R}$, it is clear that

$$\mathrm{CSW}_p(\mu, \nu)$$
$$= \left( \mathbb{E}_{K^{(1)} \sim \mathcal{U}(\mathcal{K}^{(1)}), \ldots, K^{(N)} \sim \mathcal{U}(\mathcal{K}^{(N)})} \left[ W_p^p \left( \mathcal{S}(\cdot | K^{(1)}, \ldots, K^{(N)}) \sharp \mu, \mathcal{S}(\cdot | K^{(1)}, \ldots, K^{(N)}) \sharp \nu \right) \right] \right)^{\frac{1}{p}}$$
$$\leq \max_{K^{(i)} \in \mathbb{R}^{c^{(1)} \times d^{(i)} \times d^{(i)}} \, \forall i \in [N]} W_p \left( \mathcal{S}(\cdot | K^{(1)}, \ldots, K^{(N)}) \sharp \mu, \mathcal{S}(\cdot | K^{(1)}, \ldots, K^{(N)}) \sharp \nu \right)$$
$$\leq \max_{\theta \in \mathbb{R}^{cd^2} : \|\theta\| \leq 1} W_p(\theta \sharp \mu, \theta \sharp \nu) = \text{max-SW}_p(\mu, \nu),$$

where the second inequality is due to the inequality with $\ell_2$ norm of convolution of matrices and the fact that the $\ell_2$ norm of each tensor $K^{(i)}$ is 1 for all $i \in [N]$. In addition, we find that

$$
\begin{aligned}
\text{max-SW}_p^p(\mu, \nu) &= \max_{\theta \in \mathbb{R}^{cd^2}: \|\theta\| \leq 1} \left( \inf_{\pi \in \Pi(\mu, \nu)} \int_{\mathbb{R}^{cd^2}} |\theta^\top x - \theta^\top y|^p d\pi(x, y) \right) \\
&\leq \max_{\theta \in \mathbb{R}^{cd^2}: \|\theta\| \leq 1} \left( \inf_{\pi \in \Pi(\mu, \nu)} \int_{\mathbb{R}^{cd^2} \times \mathbb{R}^{cd^2}} \|\theta\|^p \|x - y\|^p d\pi(x, y) \right) \\
&\leq \inf_{\pi \in \Pi(\mu, \nu)} \int_{\mathbb{R}^{cd^2} \mathbb{R}^{cd^2}} \|\theta\|^p \|x - y\|^p d\pi(x, y) = W_p^p(\mu, \nu).
\end{aligned}
$$

Putting the above results together, we obtain the conclusion of the proposition.

### B.3  Proof of Proposition 3

From the assumption of Proposition 3, we denote $\Theta \subset \mathbb{R}^{c \times d \times d}$ as the compact set that the probability measure $P$ is supported on. Based on the result of Proposition 2, we have

$$
\mathbb{E}[\text{CSW}_p(P_n, P)] \leq \mathbb{E}[\text{max-SW}_p(P_n, P)],
$$

where $\text{max-SW}_p(P_n, P) = \max_{\theta \in \mathbb{R}^{cd^2}: \|\theta\| \leq 1} W_p(\theta \sharp P_n, \theta \sharp P)$. Therefore, to obtain the conclusion of the proposition, it is sufficient to demonstrate that $\mathbb{E}[\text{max-SW}_p(P_n, P)] \leq C\sqrt{(cd^2 + 1) \log_2 n / n}$ for some universal constant $C > 0$. Indeed, from the closed-form expression of Wasserstein metric in one dimension, we have

$$
\begin{aligned}
\text{max-SW}_p^p(P_n, P) &= \max_{\theta \in \mathbb{R}^{cd^2}: \|\theta\| \leq 1} \int_0^1 |F_{n,\theta}^{-1}(u) - F_\theta^{-1}(u)|^p du \\
&= \max_{\theta \in \mathbb{R}^{cd^2}: \|\theta\| \leq 1} \int_{\mathbb{R}} |F_{n,\theta}(x) - F_\theta(x)|^p dx, \quad \leq \text{diam}(\Theta) \max_{\theta \in \mathbb{R}^{cd^2}: \|\theta\| \leq 1} |F_{n,\theta}(x) - F_\theta(x)|^p,
\end{aligned}
$$

where $F_{n,\theta}$ and $F_\theta$ are respectively the cumulative distributions of $\theta \sharp P_n$ and $\theta \sharp P$. Furthermore, we have the following relation:

$$
\max_{\theta \in \mathbb{R}^{cd^2}: \|\theta\| \leq 1} |F_{n,\theta}(x) - F_\theta(x)| = \sup_{A \in \mathcal{A}} |P_n(A) - P(A)|,
$$

where $\mathcal{A}$ is the set of half-spaces $\{y \in \mathbb{R}^{cd^2} : \theta^\top y \leq x\}$ for all $\theta \in \mathbb{R}^{cd^2}$ such that $\|\theta\| \leq 1$. The Vapnik-Chervonenkis (VC) dimension of $\mathcal{A}$ is upper bounded by $cd^2 + 1$ (see the reference [60]). Therefore, with probability at least $1 - \delta$ we obtain that

$$
\sup_{A \in \mathcal{A}} |P_n(A) - P(A)| \leq \sqrt{\frac{32}{n}[(cd^2 + 1) \log_2(n + 1) + \log_2(8/\delta)]}.
$$

Putting the above results together, we can conclude that $\mathbb{E}[\text{max-SW}_p(P_n, P)] \leq C\sqrt{(cd^2 + 1) \log_2 n / n}$ for some universal constant $C > 0$. As a consequence, we obtain the conclusion of the proposition.

### B.4  Proof of Proposition 1

(a) We first consider the computational and projection memory complexities of the convolution-base slicer. When $d$ is even, we can write down $d = 2^{[\log_2 d] - 1} \cdot \frac{d}{2^{[\log_2 d] - 1}}$. Direct calculation indicates

that the computational complexity of convolution-base slicer is

$$\mathcal{O}\left(\frac{d^2}{4} \cdot c\left(\frac{d}{2}+1\right)^2 + \left(\sum_{l=2}^{\lceil \log_2 d \rceil - 1}(2^{-l}d)^2(2^{-l}d+1)^2\right) + \frac{d^2}{4^{\lceil \log_2 d \rceil - 1}}\right)$$

$$= \mathcal{O}\left(\frac{cd^4}{16} + d^4\sum_{l=2}^{\lceil \log_2 d \rceil - 1}\frac{1}{16^l}\right)$$

$$= \mathcal{O}\left(\frac{cd^4}{16} - d^4 - \frac{d^4}{16} + \sum_{l=0}^{\lceil \log_2 d \rceil - 1}\frac{1}{16^l}\right)$$

$$= \frac{cd^4}{16} + \frac{d^2}{4^{\lceil \log_2 d \rceil}} - d^4 - \frac{d^4}{16} + d^4\frac{1 - \frac{1}{16^{\lceil \log_2 d \rceil}}}{1 - \frac{1}{16}}$$

$$= \mathcal{O}\left(\left(\frac{c-17}{16} + \frac{16^{\lceil \log_2 d \rceil} - 1}{15 \cdot 16^{\lceil \log_2 d \rceil - 1}}\right)d^4\right) = \mathcal{O}(cd^4).$$

Similarly, we can check that the projection memory complexity of convolution-base slicer is

$$\mathcal{O}\left(\frac{cd^2}{4} + \left(\sum_{l=2}^{\lceil \log_2 d \rceil - 1}(2^{-l}d)^2\right) + \frac{d^2}{4^{\lceil \log_2 d \rceil - 1}}\right) = \mathcal{O}\left(\frac{cd^2}{4} + \frac{d^2}{4^{\lceil \log_2 d \rceil}} - d^2 + d^2\frac{1 - \frac{1}{4^{\lceil \log_2 d \rceil}}}{1 - \frac{1}{4}}\right)$$

$$= \mathcal{O}\left(\left(\frac{c-5}{4} + \frac{4^{\lceil \log_2 d \rceil} - 1}{3 \cdot 4^{\lceil \log_2 d \rceil - 1}}\right)d^2\right) = \mathcal{O}(cd^2).$$

Therefore, we obtain the conclusion of part (a) when $d$ is even. Moving to the case when $d$ is odd, the computational complexity of convolution-base slicer becomes

$$\mathcal{O}\left(4c \cdot (d-1)^2 + \frac{(d-1)^4}{16} + \frac{(d-1)^2}{4^{\lceil \log_2(d-1) \rceil - 1}} - (d-1)^4 - \frac{(d-1)^4}{16} + (d-1)^4\frac{1 - \frac{1}{16^{\lceil \log_2(d-1) \rceil}}}{1 - \frac{1}{16}}\right)$$

$$= \mathcal{O}\left(4cd^2 + \left(\frac{16^{\lceil \log_2(d-1) \rceil} - 1}{15 \cdot 16^{\lceil \log_2(d-1) \rceil - 1}} - \frac{17}{16}\right)d^4\right) = \mathcal{O}(cd^2 + d^4).$$

Similarly, we can check that when $d$ is odd, the projection memory complexity of convolution-base slicer is $\mathcal{O}\left(4c + \left(\frac{4^{\lceil \log_2(d-1) \rceil} - 1}{3 \cdot 4^{\lceil \log_2(d-1) \rceil - 1}} - \frac{5}{4}\right)d^2\right) = \mathcal{O}(cd^2)$. As a consequence, we obtain our claims with the computational and projection memory complexities of convolution-base slicer.

(b) We now establish the computational and projection memory complexities of convolution-stride slicer. When $d$ is even, we can write down $d = 2^{\lceil \log_2 d \rceil - 1} \cdot \frac{d}{2^{\lceil \log_2 d \rceil - 1}}$. Then, the computational complexity of convolution-stride slicer is

$$\mathcal{O}\left(4c \cdot \frac{d^2}{4} + \left(\sum_{l=2}^{\lceil \log_2 d \rceil - 1}4(2^{-l}d)^2\right) + \frac{d^2}{4^{\lceil \log_2 d \rceil - 1}}\right) = \mathcal{O}\left(cd^2 + 4d^2\left(-1 - \frac{1}{4} + \frac{1 - \frac{1}{4^{\lceil \log_2 d \rceil}}}{1 - \frac{1}{4}}\right)\right)$$

$$= \mathcal{O}\left(\left(\frac{c-5}{4} + \frac{4^{\lceil \log_2 d \rceil} - 1}{3 \cdot 4^{\lceil \log_2 d \rceil - 1}}\right)d^2\right).$$

Similarly, the projection memory complexity of convolution-stride slicer is

$$\mathcal{O}\left(4c + \left(\sum_{l=2}^{\lceil \log_2 d \rceil - 1}4\right) + \frac{d^2}{4^{\lceil \log_2 d \rceil - 1}}\right) = \mathcal{O}\left(4c + \frac{d^2}{4^{\lceil \log_2 d \rceil - 1}} + 4\lceil \log_2 d \rceil\right) = \mathcal{O}(c + \lceil \log_2 d \rceil).$$

When $d$ is odd, the computational complexity of convolution-stride slicer is

$$\mathcal{O}\left(4c \cdot (d-1)^2 + 4\frac{(d-1)^2}{4} + \left(\sum_{l=2}^{[\log_2(d-1)]-1} 4(2^{-l}(d-1))^2\right) + \frac{(d-1)^2}{4^{[\log_2(d-1)]-1}}\right)$$

$$= \mathcal{O}\left(4c(d-1)^2 + 4d^2\left(-1 - \frac{1}{4} + \frac{1 - \frac{1}{4^{[\log_2(d-1)]}}}{1 - \frac{1}{4}}\right)\right)$$

$$= \mathcal{O}\left(\left(4c + \frac{4^{[\log_2(d-1)]} - 1}{3 \cdot 4^{[\log_2(d-1)]-1}} - \frac{5}{4}\right)d^2\right) := \mathcal{O}\left(cd^2\right).$$

Similarly, we can check that when $d$ is odd, the projection memory complexity of convolution-stride slicer is $\mathcal{O}\left(4c + \frac{(d-1)^2}{4^{[\log_2(d-1)]-1}} + 4[\log_2(d-1)]\right) = \mathcal{O}(c + [\log_2 d])$. As a consequence, we obtain the conclusion of part (b).

(c) Since the convolution-dilation slicer is designed in the same way as that of the convolution-stride slicer, its computational complexity and projection memory complexity can be derived in the same manner as those of the convolution-stride slicer. As a consequence, we reach the conclusion of part (c).

# C  Non-linear Versions of Convolution Sliced Wasserstein, Max Convolution Sliced Wassestein, and Convolution Projected Robust Wasserstein

In this appendix, we consider an extension of convolution sliced Wasserstein to non-linear convolution sliced Wasserstein to enhance the ability of convolution sliced Wasserstein to capture the non-linearity of the data. Moreover, we also propose the max sliced version of convolution sliced Wasserstein to overcome the projection complexity [12].

**Non-linear convolution sliced Wasserstein:** We first state the definition of non-linear convolution-base slicer.

**Definition 7** *(Non-Linear Convolution-base Slicer) Given $X \in \mathbb{R}^{c \times d \times d}$ ($d \geq 2$) and a non linear activation $\sigma(\cdot)$,*

1. *When $d$ is even, $N = [\log_2 d]$, sliced kernels are defined as $K^{(1)} \in \mathbb{R}^{1 \times 2^{-1}d+1 \times 2^{-1}d+1}$ and $K^{(h)} \in \mathbb{R}^{1 \times 2^{-h}d+1 \times 2^{-h}d+1}$ for $h = 2, \ldots, N-1$, and $K^{(N)} \in \mathbb{R}^{1 \times a \times a}$ where $a = \frac{d}{2^{N-1}}$. Then, the non-linear convolution-base slicer $\mathcal{NCS}\text{-}b(X|K^{(1)}, \ldots, K^{(N)})$ is defined as:*

$$\mathcal{NCS}\text{-}b(X|K^{(1)}, \ldots, K^{(N)}) = X^{(N)}, \quad X^{(h)} = \begin{cases} X & h = 0, \\ \sigma(X^{(h-1)} \overset{1,1}{*} K^{(h)}) & 1 \leq h \leq N-1, \\ X^{(h-1)} \overset{1,1}{*} K^{(h)} & h = N, \end{cases}$$

(3)

2. *When $d$ is odd, the non-linear convolution-base slicer $\mathcal{NCS}\text{-}b(X|K^{(1)}, \ldots, K^{(N)})$ takes the form:*

$$\mathcal{NCS}\text{-}b(X|K^{(1)}, \ldots, K^{(N)}) = \mathcal{NCS}\text{-}b(\sigma(X \overset{1,1}{*} K^{(1)})|K^{(2)}, \ldots, K^{(N)}), \quad (4)$$

*where $K^{(1)} \in \mathbb{R}^{c \times 2 \times 2}$ and $K^{(2)}, \ldots, K^{(N)}$ are the corresponding sliced kernels that are defined on the dimension $d-1$.*

The main idea of non-linear convolution-based slicer is that we incorporate non-linear activation function $\sigma(.)$ into the layers of the convolution-base slicer. Using that idea, we also can extend the convolution-stride and convolution-dilation slicers to their nonlinear versions, named non-linear convolution-stride and convolution-dilation slicers. We respectively denote these slicers as $\mathcal{NCS}\text{-}s(X|K^{(1)}, \ldots, K^{(N)})$ and $\mathcal{NCS}\text{-}d(X|K^{(1)}, \ldots, K^{(N)})$.

Using the non-linear convolution-base slicer for sliced Wasserstein, we obtain the corresponding non-linear convolution-base sliced Wasserstein as follows.

Table 2: Values of SW and CSW's variants between probability measures over digits images on MNIST with $L = 1$.

| | | 0 | 1 | 2 | 3 | 4 | 5 | 6 | 7 | 8 | 9 |
|---|---|---|---|---|---|---|---|---|---|---|---|
| 0 | SW | 0.59±0.12 | 9.4±3.33 | 8.83±5.14 | 12.34±10.13 | 14.61±8.93 | 4.43±2.4 | 10.3±5.61 | 7.89±3.39 | 10.37±7.68 | 15.92±6.76 |
| | CSW-b | 0.68±0.23 | 38.28±7.75 | 15.9±7.74 | 30.9±27.25 | 20.35±11.8 | 19.76±13.07 | 14.54±4.88 | 14.88±9.95 | 17.34±7.05 | 31.51±27.5 |
| | CSW-s | 0.42±0.22 | 18.3±11.06 | 12.57±10.71 | 13.41±12.62 | 30.13±13.13 | 8.85±4.24 | 6.8±4.31 | 7.4±5.0 | 11.24±11.9 | 25.05±22.1 |
| | CSW-d | 0.62±0.44 | 19.56±8.64 | 9.91±6.38 | 11.34±4.58 | 12.27±7.89 | 5.18±1.56 | 10.94±5.28 | 6.39±3.4 | 9.51±8.21 | 8.06±4.92 |
| 1 | SW | 18.23±12.47 | 0.32±0.08 | 8.86±3.11 | 13.46±4.29 | 10.87±4.39 | 15.77±5.76 | 11.22±8.73 | 12.69±9.66 | 9.76±2.16 | 12.43±2.42 |
| | CSW-b | 37.02±9.7 | 0.66±0.09 | 13.46±2.43 | 20.11±10.16 | 16.92±7.49 | 21.14±6.6 | 19.91±9.33 | 23.51±19.74 | 29.86±18.42 | 13.85±4.74 |
| | CSW-s | 6.33±3.27 | 0.41±0.18 | 6.93±2.68 | 7.11±1.69 | 14.36±7.01 | 13.35±7.08 | 11.82±7.18 | 7.67±4.64 | 13.43±8.93 | 9.01±4.79 |
| | CSW-d | 22.36±18.36 | 0.35±0.06 | 10.49±2.83 | 17.85±10.07 | 12.72±8.06 | 15.42±6.8 | 18.25±9.68 | 12.31±4.21 | 15.98±8.27 | 24.82±11.8 |
| 2 | SW | 8.54±7.8 | 9.24±3.79 | 0.63±0.16 | 8.73±3.34 | 13.28±7.51 | 11.86±4.23 | 12.59±6.96 | 15.69±12.09 | 9.86±4.11 | 15.02±12.31 |
| | CSW-b | 19.79±6.4 | 22.5±11.13 | 0.81±0.29 | 9.4±3.61 | 7.64±2.07 | 15.3±14.47 | 9.46±8.81 | 24.4±14.02 | 13.8±5.79 | 14.91±8.85 |
| | CSW-s | 7.05±4.92 | 13.19±5.25 | 0.6±0.16 | 5.26±3.69 | 9.01±6.03 | 18.3±13.13 | 11.09±10.71 | 7.52±5.9 | 9.66±8.72 | 10.12±13.99 |
| | CSW-d | 13.71±8.64 | 11.59±7.01 | 0.64±0.3 | 9.41±8.72 | 7.19±3.3 | 11.47±5.3 | 11.62±8.41 | 13.51±3.85 | 8.54±4.0 | 8.29±6.85 |
| 3 | SW | 10.94±10.43 | 12.5±3.35 | 7.71±6.39 | 0.56±0.09 | 6.15±4.34 | 9.42±2.55 | 8.52±4.97 | 12.61±8.26 | 16.88±11.04 | 5.58±4.21 |
| | CSW-b | 21.06±13.03 | 23.71±18.24 | 19.08±8.85 | 0.78±0.16 | 25.54±9.02 | 10.23±5.32 | 12.72±6.25 | 18.05±10.86 | 7.33±1.56 | 16.0±4.06 |
| | CSW-s | 18.19±12.12 | 16.27±14.42 | 8.06±4.73 | 0.5±0.31 | 15.18±11.24 | 4.76±2.0 | 8.88±5.27 | 9.66±6.45 | 6.99±4.95 | 8.34±9.14 |
| | CSW-d | 10.72±4.99 | 14.09±7.45 | 6.73±6.18 | 0.56±0.17 | 4.6±1.33 | 8.03±2.87 | 12.0±8.28 | 12.23±5.8 | 5.16±2.18 | 10.15±5.44 |
| 4 | SW | 16.21±10.58 | 12.17±4.06 | 12.54±10.76 | 17.58±7.57 | 0.51±0.1 | 9.57±3.96 | 7.79±4.5 | 12.73±7.48 | 11.12±3.72 | 5.6±2.48 |
| | CSW-b | 16.27±5.5 | 25.54±13.63 | 13.43±3.4 | 22.14±16.21 | 0.82±0.12 | 23.48±23.19 | 13.3±5.44 | 13.23±8.83 | 21.25±15.67 | 8.53±2.99 |
| | CSW-s | 20.57±18.68 | 14.52±10.89 | 18.37±13.61 | 12.49±7.05 | 0.47±0.25 | 9.23±9.75 | 15.37±6.19 | 7.45±7.09 | 6.73±6.33 | 5.59±2.31 |
| | CSW-d | 13.65±8.12 | 15.26±8.16 | 11.22±7.24 | 6.18±2.18 | 0.36±0.05 | 5.73±3.21 | 12.19±4.31 | 9.41±9.01 | 10.59±3.86 | 5.98±3.99 |
| 5 | SW | 12.23±5.59 | 12.99±5.15 | 17.83±11.12 | 5.3±2.56 | 8.37±2.08 | 0.58±0.08 | 4.59±3.25 | 8.8±4.54 | 5.82±2.61 | 11.44±3.38 |
| | CSW-b | 10.05±4.97 | 21.11±8.77 | 19.44±8.42 | 8.53±3.58 | 10.85±5.0 | 0.81±0.28 | 12.84±6.14 | 16.5±11.63 | 13.45±8.48 | 9.28±4.11 |
| | CSW-s | 6.85±4.73 | 8.2±5.43 | 10.48±8.54 | 16.85±18.48 | 14.13±3.52 | 0.73±0.2 | 10.42±4.81 | 5.49±3.75 | 3.82±3.15 | 10.08±5.71 |
| | CSW-d | 8.76±4.74 | 14.61±5.94 | 11.85±4.81 | 7.1±3.36 | 17.0±4.88 | 0.82±0.39 | 7.69±4.06 | 15.36±3.26 | 11.74±11.53 | 7.4±5.64 |
| 6 | SW | 16.21±9.44 | 15.84±6.29 | 6.59±1.61 | 7.94±9.3 | 6.44±2.4 | 16.24±6.96 | 0.65±0.16 | 11.23±2.59 | 17.33±10.31 | 7.3±2.58 |
| | CSW-b | 21.58±3.87 | 17.02±5.1 | 13.73±4.56 | 19.33±14.03 | 21.98±11.2 | 9.84±5.12 | 1.04±0.52 | 17.21±8.0 | 12.87±2.52 | 9.1±3.99 |
| | CSW-s | 18.79±16.51 | 14.57±8.51 | 6.2±2.06 | 14.14±11.12 | 13.41±9.32 | 10.95±10.33 | 0.71±0.09 | 12.14±9.43 | 9.59±6.6 | 6.89±4.66 |
| | CSW-d | 19.58±9.77 | 18.05±9.38 | 4.7±0.67 | 14.1±12.84 | 11.41±2.21 | 16.39±6.1 | 0.79±0.24 | 12.75±4.06 | 13.21±10.12 | 12.08±2.4 |
| 7 | SW | 10.44±4.83 | 11.62±7.83 | 8.61±5.11 | 16.65±12.8 | 9.87±5.8 | 12.64±2.6 | 14.57±4.78 | 0.47±0.14 | 9.98±3.95 | 7.49±4.21 |
| | CSW-b | 24.07±16.97 | 26.36±32.13 | 20.89±16.05 | 15.88±5.97 | 11.23±2.97 | 15.06±9.82 | 16.84±3.11 | 0.69±0.11 | 21.81±9.41 | 13.48±7.55 |
| | CSW-s | 12.37±7.59 | 12.62±11.39 | 11.9±12.84 | 12.97±7.35 | 16.3±8.65 | 4.92±2.62 | 7.9±2.57 | 0.45±0.2 | 8.66±7.18 | 4.69±5.43 |
| | CSW-d | 13.13±12.28 | 12.91±6.65 | 15.72±7.26 | 13.89±3.32 | 7.06±2.4 | 12.37±4.41 | 14.19±8.15 | 0.79±0.34 | 6.03±2.32 | 6.07±2.33 |
| 8 | SW | 11.18±3.98 | 14.19±5.37 | 6.66±3.28 | 7.15±4.18 | 7.82±3.83 | 5.76±2.75 | 20.31±12.51 | 24.66±11.34 | 0.63±0.12 | 10.91±6.09 |
| | CSW-b | 31.06±18.71 | 22.14±9.72 | 10.13±3.45 | 12.46±8.84 | 14.29±11.11 | 9.83±3.62 | 10.15±4.47 | 21.86±14.21 | 0.9±0.18 | 12.38±5.37 |
| | CSW-s | 8.43±6.04 | 15.39±12.39 | 4.16±2.58 | 5.37±2.76 | 3.35±1.86 | 4.46±2.26 | 5.44±4.0 | 15.2±11.91 | 0.56±0.19 | 7.23±3.64 |
| | CSW-d | 21.88±12.6 | 16.54±10.0 | 13.86±9.91 | 12.29±11.32 | 5.14±3.02 | 5.76±4.77 | 12.81±13.42 | 9.39±4.3 | 0.57±0.16 | 12.99±8.29 |
| 9 | SW | 18.24±10.84 | 15.09±4.64 | 9.86±6.72 | 9.79±10.32 | 5.83±5.27 | 8.39±4.23 | 9.79±6.82 | 7.97±4.03 | 9.41±4.89 | 0.58±0.11 |
| | CSW-b | 16.68±5.5 | 20.92±7.31 | 11.42±4.6 | 22.42±15.27 | 8.88±2.86 | 10.05±6.64 | 13.19±4.39 | 14.94±9.06 | 10.37±1.71 | 0.91±0.23 |
| | CSW-s | 7.66±3.52 | 10.8±8.77 | 10.83±3.42 | 8.65±3.32 | 3.43±2.52 | 6.33±5.67 | 8.23±8.12 | 7.29±3.97 | 9.77±5.89 | 0.4±0.16 |
| | CSW-d | 13.27±6.99 | 19.67±10.51 | 10.97±7.32 | 15.94±7.08 | 7.06±4.48 | 10.1±5.74 | 15.91±6.66 | 2.88±1.24 | 11.62±7.4 | 0.46±0.1 |

**Definition 8** *For any $p \geq 1$, the* non-linear convolution-base sliced Wasserstein *(NCSW-b) of order $p > 0$ between two given probability measures $\mu, \nu \in \mathcal{P}_p(\mathbb{R}^{c \times d \times d})$ is given by:*

$\mathcal{NCSW}\text{-}b_p(\mu, \nu) :=$

$$\left( \mathbb{E}_{K^{(1)} \sim \mathcal{U}(\mathcal{K}^{(1)}), \dots, K^{(N)} \sim \mathcal{U}(\mathcal{K}^{(N)})} \left[ W_p^p \left( \mathcal{NCS}\text{-}b(\cdot | K^{(1)}, \dots, K^{(N)}) \sharp \mu, \mathcal{NCS}\text{-}b(\cdot | K^{(1)}, \dots, K^{(N)}) \sharp \nu \right) \right] \right)^{\frac{1}{p}},$$
(5)

*where $\mathcal{NCS}\text{-}b(\cdot | K^{(1)}, \dots, K^{(N)})$ is a non-linear convolution-base slicer with $K^{(i)} \in \mathbb{R}^{c^{(i)} \times k^{(i)} \times k^{(i)}}$ for any $i \in [N]$ and $\mathcal{U}(\mathcal{K}^{(i)})$ is the uniform distribution with the realizations being in the set $\mathcal{K}^{(i)} = \left\{ K^{(i)} \in \mathbb{R}^{c^{(i)} \times k^{(i)} \times k^{(i)}} | \sum_{h=1}^{c^{(i)}} \sum_{i'=1}^{k^{(i)}} \sum_{j'=1}^{k^{(i)}} K_{h,i',j'}^{(i)2} = 1 \right\}.$*

By replacing the non-linear convolution-base slicer $\mathcal{NCS}\text{-}b(\cdot | K^{(1)}, \dots, K^{(N)})$ in Definition 5 by non-linear convolution-stride slicer $\mathcal{NCS}\text{-}s(\cdot | K^{(1)}, \dots, K^{(N)})$ and non-linear convolution-dilation slicer $\mathcal{NCS}\text{-}d(\cdot | K^{(1)}, \dots, K^{(N)})$, we respectively have the non-linear convolution-stride sliced Wasserstein (NCSW-s) and non-linear convolution-dilation sliced Wasserstein (NCSW-d). In Appendix D, we provide experiment results with non-linear convolution sliced Wasserstein on generative models.

**Max Convolution sliced Wasserstein:** Similar to the definition of Max-SW [12], the definition of max convolution sliced Wasserstein (Max-CSW) is as follow:

**Definition 9** *For any $p \geq 1$, the max convolution sliced Wasserstein (Max-SW) of order $p > 0$ between two given probability measures $\mu, \nu \in \mathcal{P}_p(\mathbb{R}^{c \times d \times d})$ is given by:*

$$Max\text{-}CSW(\mu, \nu) := \max_{(K^{(1)}, \dots, K^{(N)}) \in \mathcal{K}^{(1)} \times \dots \times \mathcal{K}^{(N)}} W^p \left( \mathcal{S}(\cdot | K^{(1)}, \dots, K^{(N)}) \sharp \mu, \mathcal{S}(\cdot | K^{(1)}, \dots, K^{(N)}) \sharp \nu \right),$$
(6)

where $\mathcal{S}(\cdot | K^{(1)}, \dots, K^{(N)})$ is a convolution slicer and $\mathcal{K}^1, \dots, \mathcal{K}^N$ are defined as in Definition 6. The constrained optimization in Max-CSW is solved by projected gradient ascent that is similar to

Table 3: Values of SW and CSW's variants between probability measures over digits images on MNIST with $L = 10$.

| | | 0 | 1 | 2 | 3 | 4 | 5 | 6 | 7 | 8 | 9 |
|---|---|---|---|---|---|---|---|---|---|---|---|
| 0 | SW | 0.57±0.06 | 20.53±2.52 | 15.36±2.78 | 15.74±2.2 | 18.25±1.54 | 11.42±3.99 | 14.46±1.51 | 15.8±2.52 | 15.15±1.35 | 17.48±2.0 |
| | CSW-b | 0.71±0.06 | 31.88±11.67 | 22.34±3.15 | 22.98±4.53 | 20.52±5.56 | 17.94±2.84 | 22.32±2.56 | 26.14±5.25 | 30.03±6.21 | 19.28±4.25 |
| | CSW-s | 0.58±0.06 | 20.09±5.51 | 14.48±7.14 | 13.06±3.76 | 16.45±4.3 | 13.26±2.85 | 16.7±5.72 | 20.21±5.67 | 14.91±4.91 | 16.94±8.94 |
| | CSW-d | 0.52±0.06 | 21.06±7.2 | 13.01±2.71 | 17.36±3.46 | 16.16±3.39 | 14.77±3.17 | 16.7±3.23 | 21.92±3.09 | 20.25±8.6 | 18.55±2.71 |
| 1 | SW | 25.3±7.96 | 0.43±0.03 | 16.3±1.96 | 17.36±2.74 | 16.39±2.88 | 14.01±1.93 | 19.24±3.77 | 13.23±2.57 | 15.99±1.78 | 14.52±2.62 |
| | CSW-b | 33.0±6.0 | 0.65±0.08 | 20.46±1.46 | 22.44±1.94 | 27.12±4.68 | 23.18±3.97 | 24.84±2.97 | 29.08±5.91 | 25.64±4.54 | 28.88±5.61 |
| | CSW-s | 18.97±9.36 | 0.46±0.07 | 18.06±7.38 | 16.58±4.18 | 13.58±2.54 | 12.55±1.78 | 15.62±6.23 | 15.54±4.85 | 13.74±1.98 | 14.87±3.24 |
| | CSW-d | 22.17±2.48 | 0.43±0.04 | 16.17±1.43 | 16.78±4.2 | 14.93±2.08 | 12.79±2.63 | 14.98±3.85 | 16.9±5.14 | 13.92±4.58 | 15.11±3.81 |
| 2 | SW | 14.77±1.8 | 17.69±1.6 | 0.64±0.03 | 10.28±1.96 | 12.22±2.12 | 11.73±2.48 | 11.5±3.53 | 13.53±1.72 | 9.6±1.85 | 13.15±2.57 |
| | CSW-b | 21.49±4.99 | 23.43±6.43 | 0.83±0.05 | 18.58±3.28 | 18.32±2.29 | 18.96±2.85 | 16.8±3.66 | 18.31±2.3 | 16.2±2.87 | 18.84±5.18 |
| | CSW-s | 16.89±4.26 | 17.57±2.08 | 0.63±0.07 | 11.13±3.82 | 13.88±5.17 | 12.61±5.01 | 11.15±1.74 | 14.28±2.33 | 10.19±2.03 | 16.62±4.35 |
| | CSW-d | 21.28±4.15 | 17.16±3.04 | 0.63±0.06 | 12.09±3.81 | 14.79±1.78 | 12.25±4.75 | 11.71±2.14 | 17.2±1.54 | 12.32±2.76 | 15.63±2.79 |
| 3 | SW | 15.66±4.87 | 16.82±2.5 | 14.42±1.92 | 0.6±0.07 | 13.62±1.41 | 8.05±0.6 | 15.11±2.59 | 12.19±1.27 | 10.52±2.78 | 14.35±3.08 |
| | CSW-b | 24.73±8.19 | 23.51±3.83 | 16.3±3.95 | 0.76±0.12 | 25.57±3.7 | 10.64±0.96 | 22.13±5.06 | 24.77±6.63 | 16.83±1.66 | 21.49±5.49 |
| | CSW-s | 15.61±5.91 | 15.03±5.75 | 9.41±3.99 | 0.55±0.07 | 12.78±4.56 | 8.72±3.2 | 11.83±2.8 | 14.65±4.16 | 7.58±3.0 | 13.59±1.91 |
| | CSW-d | 15.88±2.67 | 14.94±3.43 | 10.75±1.56 | 0.65±0.06 | 14.7±3.18 | 8.24±1.22 | 13.83±5.09 | 13.33±3.79 | 10.0±1.62 | 14.11±2.86 |
| 4 | SW | 18.5±1.38 | 16.94±2.19 | 12.31±3.21 | 13.48±2.23 | 0.55±0.05 | 10.39±1.66 | 13.25±2.24 | 9.44±2.86 | 11.15±2.01 | 6.83±1.49 |
| | CSW-b | 25.1±5.55 | 25.62±6.12 | 18.14±3.41 | 24.22±4.88 | 0.84±0.07 | 18.9±1.36 | 14.25±2.79 | 18.08±6.55 | 18.37±1.48 | 12.07±2.52 |
| | CSW-s | 19.48±7.65 | 15.57±5.95 | 13.02±4.05 | 15.87±1.25 | 0.55±0.13 | 11.92±1.29 | 13.8±3.54 | 10.48±2.78 | 13.51±2.76 | 6.73±1.7 |
| | CSW-d | 16.17±1.16 | 18.11±5.31 | 13.21±3.01 | 15.01±1.24 | 0.55±0.08 | 13.47±3.11 | 11.53±1.48 | 8.78±2.06 | 12.27±1.25 | 7.32±2.04 |
| 5 | SW | 11.35±2.37 | 14.34±2.0 | 11.84±1.76 | 8.13±1.63 | 10.46±0.77 | 0.62±0.07 | 8.42±0.95 | 12.71±2.66 | 7.38±0.95 | 10.03±1.81 |
| | CSW-b | 17.33±7.32 | 23.97±3.93 | 18.03±3.14 | 11.4±2.14 | 17.3±1.86 | 0.81±0.03 | 13.77±2.15 | 16.37±2.63 | 13.99±1.74 | 18.16±3.73 |
| | CSW-s | 13.45±4.62 | 13.66±2.71 | 11.13±2.89 | 8.25±1.46 | 12.8±3.12 | 0.59±0.08 | 12.31±2.51 | 13.14±2.75 | 5.94±2.38 | 10.63±3.61 |
| | CSW-d | 11.79±1.47 | 14.45±4.43 | 10.99±3.76 | 8.79±2.58 | 12.68±4.04 | 0.67±0.08 | 11.24±2.05 | 12.57±1.41 | 8.5±2.28 | 11.65±2.85 |
| 6 | SW | 15.6±1.2 | 16.65±3.04 | 10.63±1.99 | 15.93±3.09 | 12.5±1.05 | 12.58±3.18 | 0.66±0.07 | 15.37±3.21 | 11.68±2.11 | 12.12±3.93 |
| | CSW-b | 21.15±4.67 | 26.98±3.01 | 14.98±1.48 | 22.54±4.87 | 18.32±3.14 | 17.24±4.15 | 1.04±0.1 | 23.68±6.73 | 15.94±1.41 | 17.94±3.6 |
| | CSW-s | 18.83±4.56 | 16.09±4.05 | 13.72±3.78 | 15.24±4.06 | 13.0±4.2 | 17.12±4.06 | 0.64±0.08 | 17.28±4.99 | 11.69±3.35 | 11.74±2.59 |
| | CSW-d | 16.12±3.72 | 14.69±3.83 | 11.43±1.67 | 10.75±1.92 | 13.95±3.16 | 15.15±1.79 | 0.69±0.15 | 17.14±1.65 | 12.42±2.7 | 13.24±4.31 |
| 7 | SW | 18.55±2.71 | 14.24±2.65 | 14.61±1.6 | 14.12±2.33 | 11.79±2.89 | 12.15±2.79 | 17.08±1.51 | 0.72±0.09 | 12.67±2.78 | 7.98±1.63 |
| | CSW-b | 24.17±4.74 | 25.38±6.42 | 21.83±8.2 | 22.54±3.56 | 19.95±5.58 | 16.28±2.96 | 21.13±2.93 | 0.9±0.14 | 19.6±2.63 | 12.73±3.11 |
| | CSW-s | 12.47±1.5 | 15.36±0.97 | 15.23±3.91 | 12.71±1.72 | 10.69±3.32 | 12.01±5.21 | 17.81±5.83 | 0.61±0.06 | 13.33±6.67 | 9.56±2.85 |
| | CSW-d | 19.4±4.62 | 17.74±3.06 | 15.3±3.29 | 10.51±2.69 | 12.01±2.19 | 11.87±2.01 | 15.7±3.37 | 0.7±0.1 | 12.91±1.35 | 9.49±2.61 |
| 8 | SW | 14.99±1.95 | 13.63±2.87 | 9.59±2.8 | 8.77±1.09 | 11.89±2.73 | 7.5±1.87 | 12.93±2.57 | 13.43±0.92 | 0.59±0.07 | 11.0±1.95 |
| | CSW-b | 24.82±3.3 | 20.56±1.82 | 15.07±2.03 | 15.51±1.18 | 18.82±3.58 | 11.73±0.52 | 16.3±3.85 | 19.28±2.76 | 0.93±0.16 | 13.9±3.14 |
| | CSW-s | 15.49±4.93 | 13.59±3.22 | 12.38±2.73 | 9.3±0.57 | 13.71±2.3 | 7.81±3.45 | 17.73±7.39 | 12.52±4.11 | 0.67±0.09 | 11.11±2.33 |
| | CSW-d | 15.19±4.72 | 13.92±2.28 | 11.3±2.84 | 10.36±2.18 | 13.92±3.61 | 8.26±1.93 | 11.13±2.68 | 13.94±1.75 | 0.61±0.07 | 10.63±2.94 |
| 9 | SW | 18.69±3.5 | 15.59±2.36 | 13.37±0.4 | 12.71±2.73 | 7.36±1.82 | 10.05±2.31 | 13.42±2.92 | 8.5±2.18 | 11.33±1.33 | 0.61±0.07 |
| | CSW-b | 25.66±7.64 | 24.44±1.89 | 20.66±6.8 | 22.19±6.0 | 9.87±1.96 | 15.43±1.32 | 16.71±4.28 | 15.41±2.76 | 15.67±2.73 | 0.8±0.12 |
| | CSW-s | 15.6±3.63 | 19.29±5.63 | 10.75±3.21 | 14.83±3.5 | 8.66±2.2 | 10.49±2.57 | 13.57±2.71 | 7.91±2.74 | 11.98±3.98 | 0.61±0.08 |
| | CSW-d | 18.11±2.98 | 15.13±3.83 | 14.29±2.38 | 13.52±3.24 | 7.41±1.78 | 10.48±0.75 | 11.89±1.63 | 11.17±3.27 | 11.17±2.6 | 0.54±0.08 |

**Max-SW.** Similar to CSW, Max-CSW also has three variants that are corresponding to three types of proposed convolution slicer, namely, Max-CSW-b, Max-CSW-s, and Max-CSW-d.

**Convolution projected robust Wasserstein:** As a generalization of Max-SW, projected robust Wasserstein (PRW) [48] finds the best subspace of $k > 1$ dimension that can maximize the Wasserstein distance between projected measures. Given two probability measures $\mu, \nu \in \mathcal{P}_p(\mathbb{R}^d)$, the projected robust Wasserstein distance between $\mu$ and $\nu$ is defined as:

$$PRW_k(\mu, \nu) := \max_{U \in \mathbb{V}_k(\mathbb{R}^d)} W_p(U \sharp \mu, U \sharp \nu), \tag{7}$$

where $\mathbb{V}_k(\mathbb{R}^d) := \{U \in \mathbb{R}^{d \times k} | U^\top U = I_k\}$ is the Stefel Manifold.

To define the convolution projected robust Wasserstein, we first define the $k$-convolution slicers:

**Definition 10** (*k-Convolution Slicer*) *For* $N \geq 1$, *given a sequence of kernels* $K^{(1)} \in \mathbb{R}^{c^{(1)} \times d^{(1)} \times d^{(1)}}, \ldots, K^{(N)} \in \mathbb{R}^{c^{(N)} \times d^{(N)} \times d^{(N)}}$, *a* $k$-convolution slicer $\mathcal{S}_k(\cdot | K^{(1)}, \ldots, K^{(N)})$ *on* $\mathbb{R}^{c \times d \times d}$ *is a composition of* $N$ *convolution functions with kernels* $K^{(1)}, \ldots, K^{(N)}$ *(with stride or dilation if needed) such that* $\mathcal{S}_k(X | K^{(1)}, \ldots, K^{(N)}) \in \mathbb{R}^k$ $\forall X \in \mathbb{R}^{c \times d \times d}$.

From the above definition, we can define the convolution projected robust Wasserstein as follow:

**Definition 11** *For any* $p \geq 1$, *the convolution projected sliced Wasserstein (CPRW) of order* $p > 0$ *between two given probability measures* $\mu, \nu \in \mathcal{P}_p(\mathbb{R}^{c \times d \times d})$ *is given by:*

$$CPRW_k(\mu, \nu) := \max_{(K^{(1)}, \ldots, K^{(N)}) \in \mathcal{K}^{(1)} \times \ldots \times \mathcal{K}^{(N)}} W^p \left( \mathcal{S}_k(\cdot | K^{(1)}, \ldots, K^{(N)}) \sharp \mu, \mathcal{S}_k(\cdot | K^{(1)}, \ldots, K^{(N)}) \sharp \nu \right),$$
$$(8)$$

where $\mathcal{S}_k(\cdot | K^{(1)}, \ldots, K^{(N)})$ is a $k$-convolution slicer and $\mathcal{K}^1, \ldots, \mathcal{K}^N$ are defined as in Definition 6. We can obtain three instances of $k$-convolution slicers by modifying the number of channels from 1

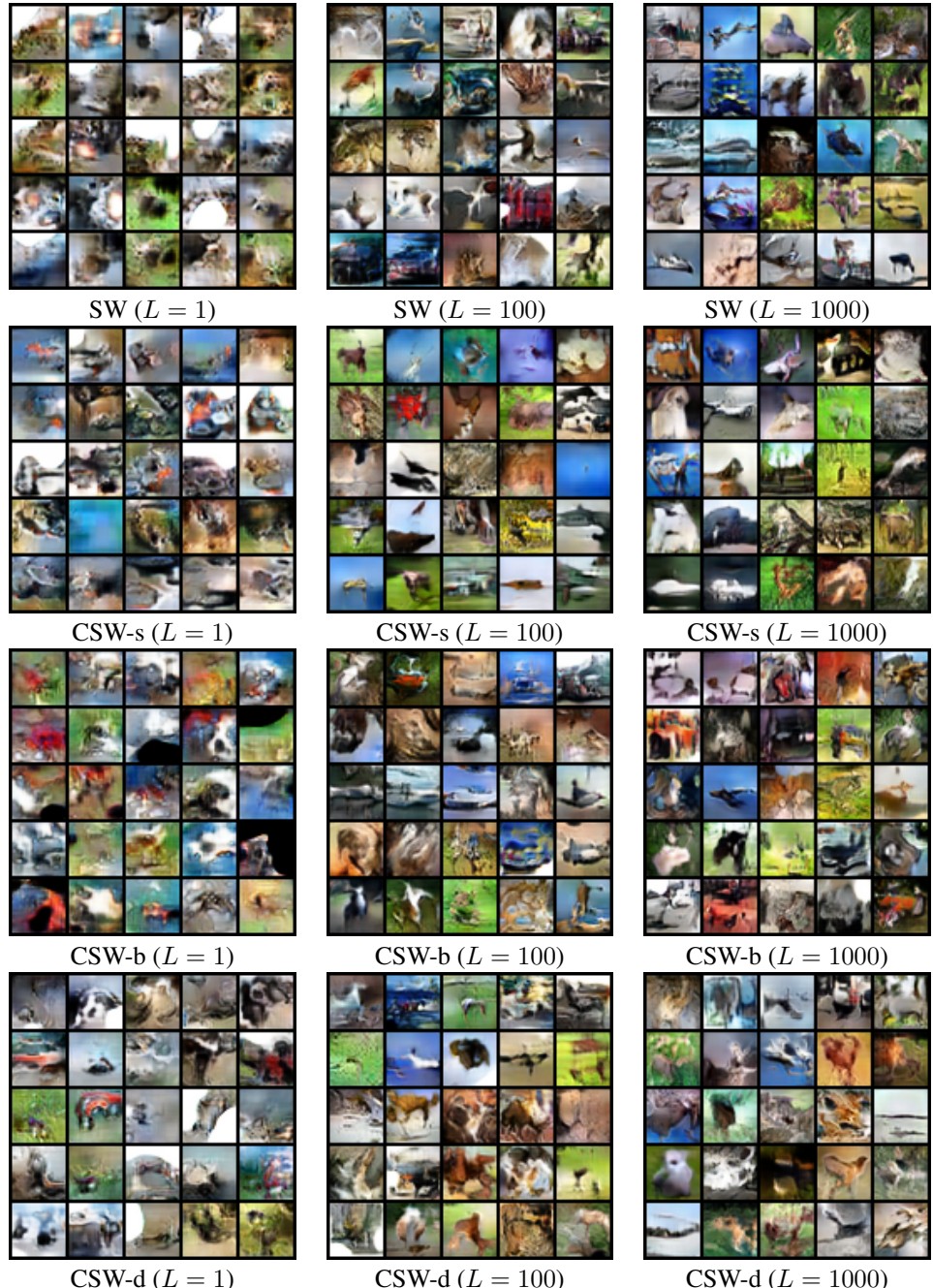

Figure 5: Random generated images of SW, CSW-s, CSW-b and CSW-d on CIFAR10.

to $k$ in the convolution-base slicer, the convolution-stride slicer, and the convolution-dilation slicer. As a result, we obtain three variants of CPRW which are CPRW-b, CPRW-s, and CPRW-d.

# D  Additional Experiments

In this section, we first present experiments on comparing probability measures over MNIST's digits in Appendix D.1. Then, we provide details of training generative models and additional experimental results in Appendix D.2.

## D.1 Comparing Measures over MNIST's digits

In the MNIST dataset, there are 60000 images of size $28 \times 28$ of digits from 0 to 9. We compute SW between two empirical probability measures over images of every two digits, e.g., 1 and 2, 1 and 3, and so on. To compare on the same digit, e,g, 1, we split images of the same digit into two disjoint sets and then compute the SW between the corresponding empirical probability measures.

Table 4: Values of SW and CSW variants between probability measures over digits images on MNIST with $L = 100$

| | | 0 | 1 | 2 | 3 | 4 | 5 | 6 | 7 | 8 | 9 |
|---|---|---|---|---|---|---|---|---|---|---|---|
| 0 | SW | 0.58±0.01 | 23.19±0.88 | 15.81±0.88 | 15.31±0.83 | 17.25±0.57 | 12.45±0.91 | 16.44±0.8 | 17.71±0.71 | 15.8±1.12 | 18.14±0.94 |
| | CSW-b | 0.83±0.03 | 32.33±3.02 | 24.86±2.11 | 25.73±2.43 | 24.71±2.55 | 18.6±1.76 | 21.86±1.71 | 25.6±1.72 | 27.24±2.36 | 24.93±0.92 |
| | CSW-s | 0.59±0.04 | 24.13±2.36 | 16.95±1.21 | 15.21±2.02 | 19.2±1.33 | 13.33±1.85 | 18.0±1.57 | 18.04±2.21 | 15.51±2.21 | 17.99±2.64 |
| | CSW-d | 0.59±0.01 | 22.65±1.47 | 16.15±1.28 | 16.79±0.79 | 17.91±0.65 | 12.6±1.28 | 17.81±1.28 | 18.53±1.54 | 14.85±1.76 | 16.93±0.97 |
| 1 | SW | 22.36±0.92 | 0.45±0.0 | 16.48±1.24 | 16.26±0.48 | 16.58±0.79 | 15.53±0.37 | 16.95±1.04 | 15.71±0.8 | 14.59±0.45 | 15.82±0.67 |
| | CSW-b | 34.71±1.82 | 0.65±0.02 | 24.19±2.05 | 25.62±1.61 | 27.75±1.6 | 23.7±1.92 | 28.07±0.58 | 27.05±2.75 | 23.84±1.37 | 25.44±0.93 |
| | CSW-s | 22.59±3.07 | 0.45±0.03 | 16.04±1.25 | 17.2±0.8 | 16.25±1.13 | 15.7±1.3 | 17.37±1.37 | 15.87±0.76 | 15.85±0.96 | 17.08±0.96 |
| | CSW-d | 23.48±1.47 | 0.46±0.01 | 16.41±0.73 | 16.39±0.74 | 16.93±0.99 | 15.01±0.74 | 16.85±1.02 | 16.48±0.99 | 15.22±0.78 | 15.76±0.8 |
| 2 | SW | 16.03±0.84 | 16.4±0.29 | 0.62±0.02 | 12.9±0.53 | 12.98±1.39 | 12.83±0.39 | 11.11±0.31 | 16.41±0.54 | 11.35±0.79 | 14.61±0.75 |
| | CSW-b | 24.7±0.84 | 24.57±1.05 | 0.89±0.05 | 19.56±1.07 | 19.09±0.48 | 20.65±1.91 | 17.95±0.94 | 20.9±1.96 | 16.98±1.21 | 18.81±0.66 |
| | CSW-s | 16.38±1.76 | 16.3±0.87 | 0.64±0.03 | 11.92±0.89 | 14.81±2.17 | 11.42±1.09 | 11.3±0.85 | 15.27±1.29 | 10.58±1.38 | 14.84±2.31 |
| | CSW-d | 16.22±0.98 | 17.09±0.93 | 0.6±0.01 | 13.22±0.37 | 13.81±0.73 | 11.92±0.5 | 12.13±1.0 | 16.3±0.93 | 11.82±1.26 | 15.26±1.45 |
| 3 | SW | 15.89±0.82 | 15.7±0.63 | 12.6±0.96 | 0.57±0.01 | 15.04±0.93 | 8.89±0.57 | 14.96±1.34 | 14.8±0.46 | 9.85±0.62 | 13.52±0.77 |
| | CSW-b | 26.62±1.65 | 25.43±3.13 | 18.57±1.66 | 0.87±0.05 | 22.38±2.45 | 14.11±1.52 | 23.83±2.36 | 24.15±1.44 | 17.0±1.84 | 19.68±1.21 |
| | CSW-s | 16.71±1.88 | 16.25±1.41 | 12.31±1.55 | 0.6±0.01 | 13.7±0.91 | 8.97±1.41 | 15.69±1.04 | 14.94±1.41 | 10.91±0.63 | 14.07±1.26 |
| | CSW-d | 15.23±1.83 | 16.37±1.05 | 13.19±0.79 | 0.58±0.02 | 15.0±0.91 | 9.21±0.61 | 16.14±0.32 | 15.64±1.24 | 11.1±0.76 | 13.93±0.6 |
| 4 | SW | 17.02±1.0 | 16.82±0.86 | 12.61±0.55 | 14.75±0.99 | 0.58±0.01 | 11.39±0.44 | 12.07±0.51 | 10.51±0.56 | 12.43±0.78 | 6.78±0.47 |
| | CSW-b | 26.86±2.04 | 26.44±1.75 | 18.91±2.74 | 22.08±1.47 | 0.83±0.03 | 18.51±1.15 | 18.49±1.35 | 18.95±1.67 | 17.29±2.19 | 10.54±0.69 |
| | CSW-s | 16.2±2.1 | 15.65±1.16 | 13.94±1.92 | 15.23±1.32 | 0.58±0.03 | 11.29±2.18 | 12.33±1.05 | 11.07±0.9 | 12.39±1.71 | 7.84±0.79 |
| | CSW-d | 17.34±1.77 | 17.28±1.27 | 13.08±1.54 | 15.3±0.67 | 0.57±0.01 | 12.0±0.52 | 13.23±0.44 | 11.98±0.71 | 11.39±0.75 | 7.26±0.51 |
| 5 | SW | 11.77±0.36 | 14.55±0.93 | 12.64±0.47 | 8.7±0.71 | 10.68±1.3 | 0.64±0.01 | 11.83±0.83 | 12.54±0.2 | 8.99±0.78 | 10.4±0.75 |
| | CSW-b | 20.55±1.98 | 25.31±2.14 | 19.68±0.92 | 13.55±1.5 | 18.43±1.22 | 0.91±0.02 | 16.55±1.0 | 17.45±0.8 | 14.4±1.07 | 15.85±1.21 |
| | CSW-s | 13.04±0.61 | 15.15±1.18 | 12.65±0.94 | 8.27±1.01 | 11.83±0.85 | 0.62±0.01 | 12.04±1.0 | 12.36±1.48 | 8.64±0.55 | 10.8±0.97 |
| | CSW-d | 11.79±1.28 | 15.31±1.15 | 13.54±1.22 | 8.82±1.07 | 12.33±0.75 | 0.62±0.04 | 12.45±0.79 | 13.02±0.81 | 9.18±0.54 | 10.73±0.85 |
| 6 | SW | 15.97±0.87 | 16.84±1.4 | 11.52±0.53 | 15.56±0.66 | 12.09±0.63 | 11.98±0.82 | 0.65±0.01 | 16.69±1.63 | 12.52±0.42 | 13.84±0.93 |
| | CSW-b | 25.66±2.37 | 26.39±0.68 | 15.93±0.91 | 22.98±3.47 | 18.8±1.9 | 17.0±1.66 | 0.91±0.02 | 23.31±2.45 | 17.62±0.99 | 18.73±0.84 |
| | CSW-s | 17.84±1.85 | 17.61±1.92 | 11.49±0.42 | 14.07±1.43 | 12.25±1.23 | 11.74±0.35 | 0.66±0.01 | 15.71±1.03 | 13.33±0.68 | 12.55±1.4 |
| | CSW-d | 16.95±1.45 | 17.15±1.12 | 11.47±0.79 | 15.71±1.24 | 11.91±0.37 | 12.63±0.94 | 0.67±0.02 | 16.36±1.29 | 13.15±1.0 | 14.35±0.92 |
| 7 | SW | 17.55±1.35 | 16.65±0.79 | 15.3±0.83 | 15.47±0.73 | 11.39±0.77 | 12.4±0.54 | 16.04±1.19 | 0.61±0.01 | 13.66±1.12 | 8.16±0.06 |
| | CSW-b | 27.36±2.07 | 28.35±1.32 | 22.24±1.59 | 23.56±1.2 | 18.46±2.75 | 19.32±1.68 | 25.38±1.94 | 0.94±0.04 | 22.63±1.67 | 14.71±0.52 |
| | CSW-s | 16.74±2.14 | 15.81±1.23 | 17.72±2.26 | 14.75±0.83 | 13.6±2.24 | 13.45±1.94 | 15.37±1.44 | 0.64±0.05 | 12.92±0.77 | 8.95±1.3 |
| | CSW-d | 18.21±1.44 | 16.31±1.55 | 16.3±1.05 | 14.97±0.76 | 11.45±0.35 | 12.82±1.54 | 16.9±0.95 | 0.69±0.04 | 13.3±0.59 | 8.72±0.48 |
| 8 | SW | 16.16±1.14 | 15.09±0.96 | 11.02±0.54 | 10.02±0.79 | 11.45±0.69 | 8.46±0.75 | 13.41±0.29 | 14.33±1.27 | 0.65±0.02 | 10.62±0.35 |
| | CSW-b | 26.49±2.12 | 21.76±0.63 | 15.73±1.07 | 17.16±1.58 | 18.25±1.36 | 14.5±0.94 | 18.87±1.68 | 21.36±1.76 | 0.97±0.04 | 15.85±0.81 |
| | CSW-s | 17.19±1.17 | 14.26±1.07 | 11.01±0.79 | 10.32±1.02 | 11.86±1.4 | 8.75±0.63 | 13.23±0.96 | 13.72±1.3 | 0.66±0.04 | 10.65±1.02 |
| | CSW-d | 15.42±1.31 | 15.41±0.87 | 11.06±0.43 | 10.56±0.44 | 12.51±1.74 | 8.98±0.61 | 13.87±1.29 | 14.77±0.67 | 0.65±0.03 | 11.09±1.06 |
| 9 | SW | 17.94±1.19 | 15.68±0.64 | 13.83±1.05 | 12.72±0.48 | 7.37±0.66 | 10.62±0.92 | 13.54±0.48 | 8.24±0.31 | 10.66±0.38 | 0.59±0.02 |
| | CSW-b | 26.67±3.65 | 26.0±1.95 | 20.52±1.24 | 19.68±1.14 | 10.39±0.42 | 16.36±2.03 | 19.24±0.99 | 14.95±1.29 | 15.71±1.44 | 0.84±0.04 |
| | CSW-s | 16.73±1.84 | 16.04±1.28 | 14.31±1.66 | 13.22±1.43 | 7.42±0.45 | 10.32±0.65 | 13.74±2.08 | 8.64±0.8 | 10.52±1.33 | 0.6±0.02 |
| | CSW-d | 17.58±1.17 | 15.43±1.09 | 13.98±0.51 | 13.55±1.43 | 7.18±0.37 | 10.89±0.58 | 13.94±1.11 | 8.58±0.42 | 11.68±0.62 | 0.6±0.02 |

**Meaningful measures of discrepancy:** We approximate the SW and the CSW's variants by a finite number of projections, namely, $L = 1$, $L = 10$, and $L = 100$. We show the mean of approximated values of $L = 100$ over 5 different runs and the corresponding standard deviation in Table 4. According to the table, we observe that SW and CSW's variants can preserve discrepancy between digits well. In particular, the discrepancies between probability measures of the same digit are relatively small compared to the discrepancies between probability measures of different digits. Moreover, we see that the values of CSW-s and CSW-d are closed to the values of SW on the same pairs of digits. We also show similar tables for $L = 1$ and $L = 10$ in Tables 2-3. From these tables, we observe that the number of projections can affect the stability of both SW and CSW's variants. Furthermore, with a smaller value of $L$, the standard deviations of 5 different runs of both SW and CSW's variants are higher than values with $L = 100$.

**Projection memory for slicers:** For SW, the conventional slicing requires $L \cdot 784$ float variables for $L$ projecting directions of $28 \cdot 28$ dimension. On the other hand, CSW only needs $L \cdot 338$ float variables since each projecting direction is represented as three kernels $K^{(1)} \in \mathbb{R}^{15 \times 15}$, $K^{(2)} \in \mathbb{R}^{8 \times 8}$, and $K^{(3)} \in \mathbb{R}^{7 \times 7}$. More importantly, CSW-s and CSW-d require only $L \cdot 57$ float variables since they are represented by three kernels $K^{(1)} \in \mathbb{R}^{2 \times 2}$, $K^{(2)} \in \mathbb{R}^{2 \times 2}$, and $K^{(3)} \in \mathbb{R}^{7 \times 7}$. From this experiment, we can see that using the whole unit-hypersphere as the space of projecting directions can be sub-optimal when dealing with images.

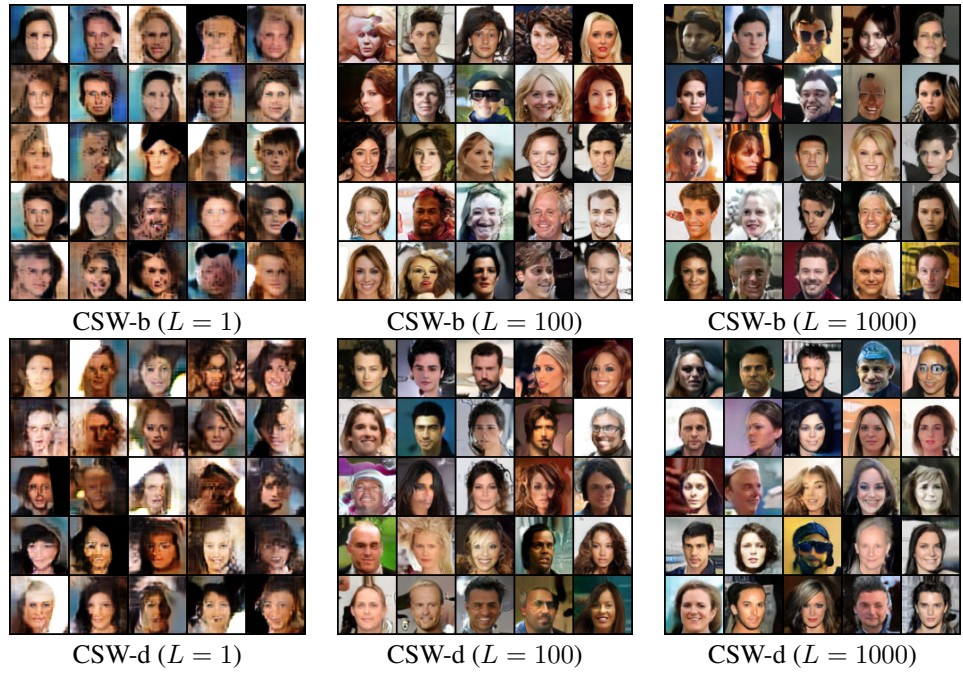

| CSW-b ($L = 1$) | CSW-b ($L = 100$) | CSW-b ($L = 1000$) |

| CSW-d ($L = 1$) | CSW-d ($L = 100$) | CSW-d ($L = 1000$) |

Figure 6: Random generated images of CSW-b and CSW-d on CelebA.

Table 5: Computational time and memory of methods (in iterations per a second and megabytes (MB).

| Method | CIFAR10 (32x32) | | CelebA (64x64) | | STL10 (96x96) | | CelebA-HQ (128x128) | |
|---|---|---|---|---|---|---|---|---|
| | Iters/s ($\uparrow$) | Mem ($\downarrow$) | Iters/s ($\uparrow$) | Mem ($\downarrow$) | Iters/s ($\uparrow$) | Mem ($\downarrow$) | Iters/s ($\uparrow$) | Mem ($\downarrow$) |
| SW (L=1) | 18.98 | 2071 | 6.21 | 8003 | 9.59 | 4596 | 10.35 | 4109 |
| SW (L=100) | 18.53 | 2080 | 6.16 | 8015 | 9.47 | 4601 | 10.22 | 4117 |
| SW (L=1000) | 18.15 | 2169 | 6.10 | 8102 | 9.13 | 4647 | 10.17 | 4202 |
| CSW-b (L=1) | 18.43 | 2070 | 6.21 | 8003 | 9.56 | 4596 | 10.33 | 4109 |
| CSW-b (L=100) | 18.35 | 2077 | 6.15 | 8009 | 9.40 | 4598 | 10.19 | 4110 |
| CSW-b (L=1000) | 18.06 | 2117 | 6.10 | 8049 | 9.07 | 4613 | 10.12 | 4134 |
| CSW-s (d) (L=1) | 18.69 | 2070 | 6.21 | 8003 | 9.56 | 4596 | 10.33 | 4109 |
| CSW-s (d) (L=100) | 18.50 | 2073 | 6.16 | 8005 | 9.41 | 4597 | 10.20 | 4109 |
| CSW-s (d) (L=1000) | 18.10 | 2098 | 6.10 | 8029 | 9.10 | 4603 | 10.12 | 4114 |

## D.2 Generative models

We parameterize the model distribution $p_\phi(x) \in \mathcal{P}(\mathbb{R}^{c \times d \times d})$ and $p_\phi(x) = G_\phi \sharp \epsilon$ where $\epsilon$ is the standard multivariate Gaussian of 128 dimension and $G_\phi$ is a neural network with Resnet architecture [19]. Since the ground truth metric between images is unknown, we need a discriminator as a type of ground metric learning. We denote the discriminator as a function $T_{\beta_2} \circ T_{\beta_1}$ where $T_{\beta_1} : \mathbb{R}^{c \times d \times d} \to \mathbb{R}^{c' \times d' \times d'}$ and $T_{\beta_2} : \mathbb{R}^{c' \times d' \times d'} \to \mathbb{R}$. In greater detail, $T_{\beta_1}$ maps the original images to their corresponding features maps and $T_{\beta_2}$ maps their features maps to their corresponding discriminative scores. Let the data distribution is $\mu$, our training objectives are:

$$\min_{\beta_1, \beta_2} \left( \mathbb{E}_{x \sim \mu}[\min(0, -1 + T_{\beta_2}(T_{\beta_1}(x)))] + \mathbb{E}_{z \sim \epsilon}[\min(0, -1 - T_{\beta_2}(T_{\beta_1}(G_\phi(z))))] \right),$$

$$\min_\phi \mathbb{E}_{X \sim \mu^{\otimes m}, Y \sim \epsilon^{\otimes m}} \mathcal{D}(T_{\beta_1} \sharp P_X, T_{\beta_1} \sharp G_\phi \sharp P_Y),$$

where $m \geq 1$ is the mini-batch size and $\mathcal{D}(\cdot, \cdot)$ is the SW or CSW's variants. The above training procedure follows the papers [13, 44] that can be seen as an application of mini-batch optimal transport [14, 46, 45] with sliced Wasserstein kernels.

**Training time and training memory:** We report in Table 5 the training speed in the number of iterations per second and the training memory in megabytes (MBs). We would like to recall that the time complexity and the projection memory complexity of CSW-s and CSW-d are the same. Therefore, we measure the training time and the training memory of CSW-s as the result for both CSW-s and CSW-d. We can see that increasing the number of projections $L$ costs more memory and

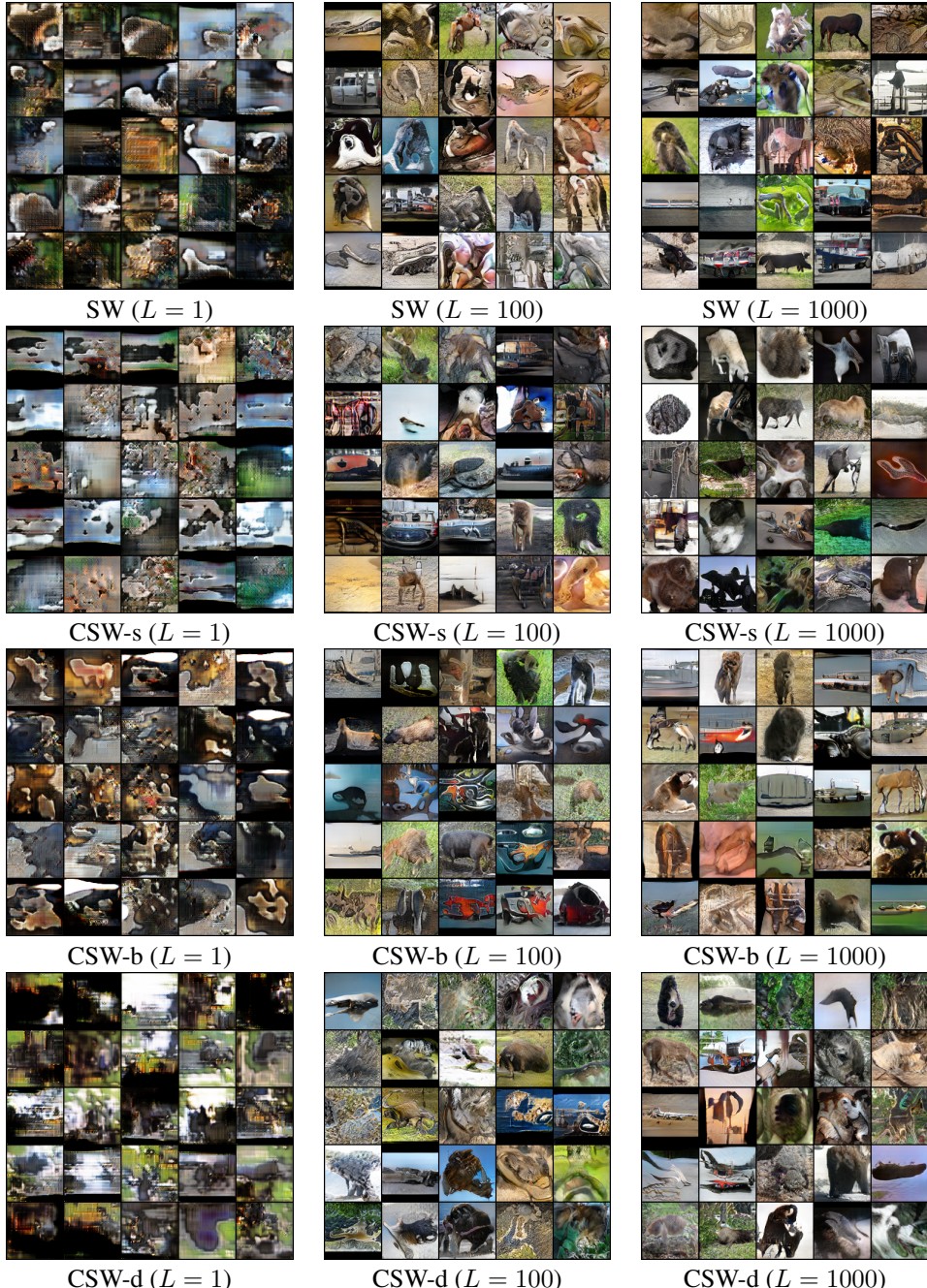

Figure 7: Random generated images of SW, CSW-s, CSW-b, and CSW-d on STL10.

also slows down the training speed. However, the rate of increasing memory of CSW is smaller than SW. For CSW-s and CSW-d, the extent of saving memory is even better. As an example, $L = 1000$ in CSW-s and CSW-d costs less memory than SW with $L = 100$ while the performance is better (see Table 1). In terms of training time, CSW-s and CSW-d are comparable to SW and they can be computed faster than CSW. We refer the readers to Section 3 for a detailed discussion about the computational time and projection memory complexity of CSW's variants.

**Random generated images:** We show some images that are drawn randomly from models trained by SW, CSW-b, CSW-s, and CSW-d on CIFAR10. CelebA, STL10, and CelebA-HQ in Figure 5, Figure 5, Figure 6, Figure 7, and Figure 8 in turn. From these figures, we again observe the effect of changing the number of projections $L$, namely, a bigger value of $L$ leads to better-generated images.

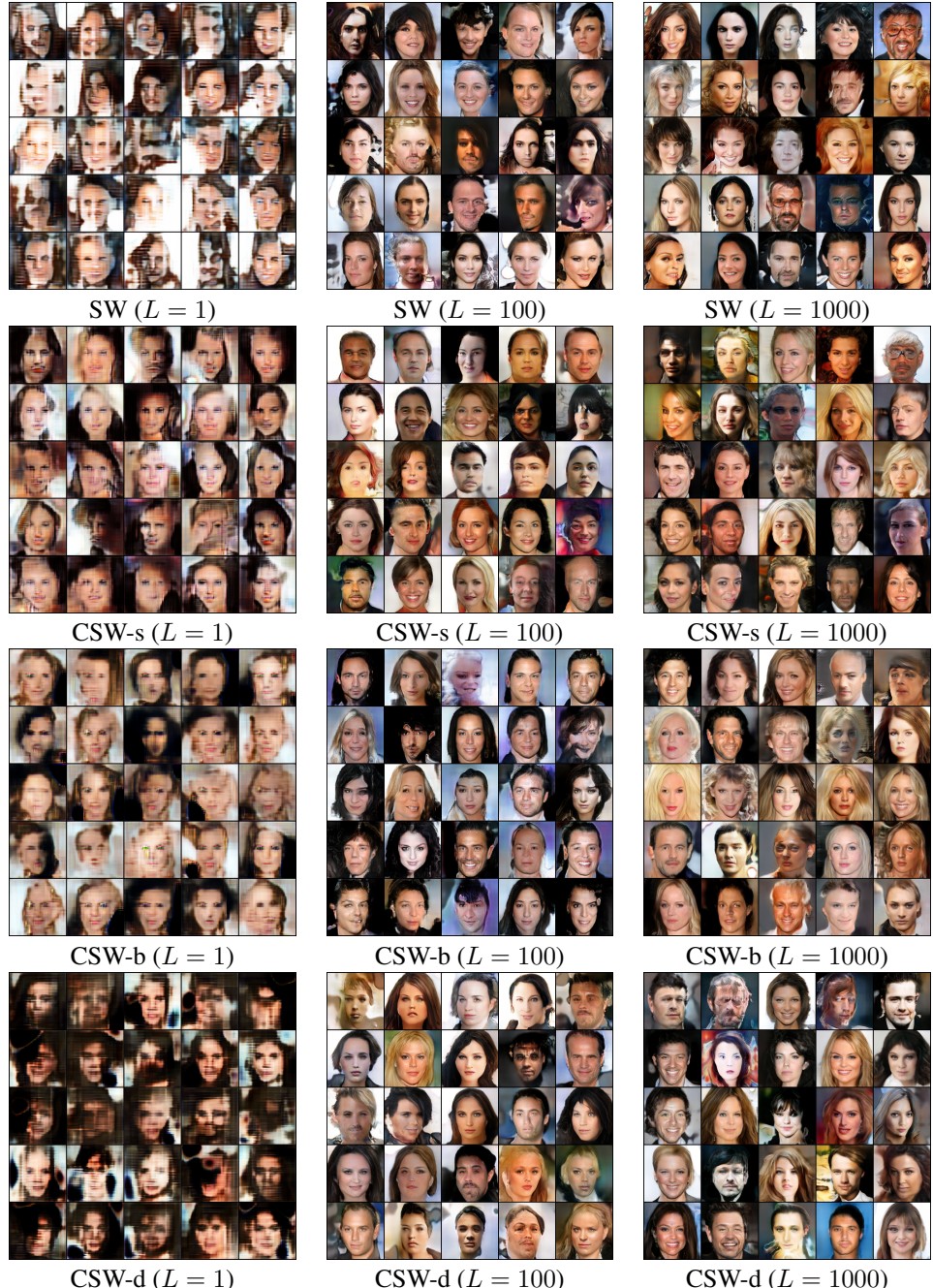

Figure 8: Random generated images of SW, CSW-s, CSW-b, and CSW-d on CelebA-HQ.

Moreover, we observe that convolution sliced Wasserstein variants provide more realistic images than the conventional sliced Wasserstein. These qualitative comparisons are consistent with the quantitative comparison via the FID scores and the IS scores in Table 1.

**Results of Max Convolution sliced Wasserstein:** We train generative models with Max-SW and Max-CSW variants. We search for the best learning rate in $\{0.1, 0.01\}$ and the number of update steps in $\{10, 100\}$. We report the best results on CIFAR10, CelebA, and CelebA-HQ for all models in Table 6. From this table, we observe that Max-CSW-s gives the best result on CIFAR10 and CelebA while Max-CSW-d is the best on CelebA-HQ. This strengthens the claim that convolution slicers are better than conventional ones. We also would like to recall that the computational time and memory of Max-CSW variants are better than Max-SW.

Table 6: Summary of FID and IS scores of Max-SW and Max-CSW variants on CIFAR10 (32x32), CelebA (64x64), and CelebA-HQ (128x128).

| Method | CIFAR10 (32x32) | | CelebA (64x64) | CelebA-HQ (128x128) |
|---|---|---|---|---|
| | FID ($\downarrow$) | IS ($\uparrow$) | FID ($\downarrow$) | FID ($\downarrow$) |
| Max-SW | 43.33 | 5.79 | 16.79 | 39.75 |
| Max-CSW-b | 44.17 | 6.19 | 14.28 | 57.70 |
| Max-CSW-s | **41.88** | 6.38 | **11.83** | 40.84 |
| Max-CSW-d | 44.21 | **6.42** | 12.06 | **39.17** |
| PRW (k=2) | 44.74 | 6.00 | | |
| CPRW-s (k=2) | **37.61** | **6.53** | | |
| PRW (k=4) | 41.39 | 6.10 | | |
| CPRW-s (k=4) | **40.07** | **6.30** | | |
| PRW (k=16) | 39.51 | 6.38 | | |
| CPRW-s (k=4) | **38.22** | **6.45** | | |

Table 7: Summary of FID and IS scores of NSW and NCSW variants on CIFAR10 (32x32).

| Method | CIFAR10 (32x32) | |
|---|---|---|
| | FID ($\downarrow$) | IS ($\uparrow$) |
| NSW (L=1) | 83.58 | 3.76 |
| NCSW-b (L=1) | 82.19 | 3.74 |
| NCSW-s (L=1) | 79.09 | **4.42** |
| NCSW-d (L=1) | **75.9**4 | 3.92 |
| NSW (L=100) | 52.99 | 5.33 |
| NCSW-b (L=100) | 50.25 | 5.60 |
| NCSW-s (L=100) | **44.56** | 5.91 |
| NCSW-d (L=100) | 45.91 | **6.04** |
| NSW (L=1000) | 43.73 | 6.03 |
| NCSW-b (L=1000) | 44.03 | 5.98 |
| NCSW-s (L=1000) | **30.21** | **6.97** |
| NCSW-d (L=1000) | 42.30 | 6.31 |

**Results of Convolution projected sliced Wasserstein:** As generalization of Max-SW and Max-CSW, we use PRW and CPRW-s with $k \in \{2, 4, 16\}$ to train generative models. We search for the best learning rate in $\{0.1, 0.01\}$ and the number of update steps in $\{10, 100\}$. The result on CIFAR is given in Table 6. According to the table, CPRW-s is better than PRW with all choice of $k$ which reinforces the favorable performance of convolution slicers.

**Results of non-linear convolution sliced Wasserstein:** We report FID scores and IS scores of generative models trained by non-linear sliced Wasserstein (NSW) [24] and non-linear convolution sliced Wasserstein (NCSW) variants including NCSW-b, NCSW-s, and NCSW-d on CIFAR10 in Table 7. The non-linear sliced Wasserstein is a variant of generalized sliced Wasserstein where we use a non-linear activation function after the linear projection, namely, $g(x, \theta) = \sigma(\theta^\top x)$. For NSW and NCSW variants, we choose $\sigma()$ as the Sigmoid function. Compared to linear versions in Table 1, we can see that including the non-linear activation function can improve the scores in some cases, e.g., NSW and NCSW-s. We also show FID scores and IS scores across training epochs in Figure 9. Similar to the linear case, NCSW's variants can help generative models converge faster than NSW.

# E Experimental Settings

**Architectures of neural network:** We illustrate the detail of neural network architectures including the generative networks and the discriminative networks on CIFAR10 in Table 8, CelebA in Table 9, STL10 in Table 10, and CelebA-HQ in Table 11.

**Other settings:** We set the number of training iterations to 50000 on CIFAR10, CelebA, and CelebA-HQ and to 100000 on STL10. For each 5 iterations, we update the generator $G_\phi$ by the

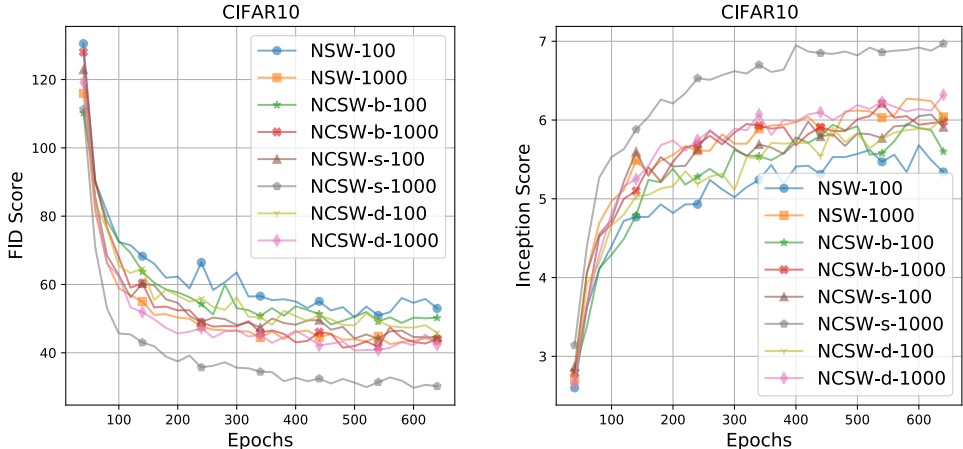

Figure 9: FID scores and IS scores over epochs of different training non-linear losses on CIFAR10. We observe that NCSW variants usually help the generative models converge faster.

Table 8: CIFAR10 architectures.

| (a) $G_\phi$ |
| --- |
| Input: $\boldsymbol{\epsilon} \in \mathbb{R}^{128} \sim \mathcal{N}(0,1)$ |
| $128 \to 4 \times 4 \times 256$, dense linear |
| ResBlock up 256 |
| ResBlock up 256 |
| ResBlock up 256 |
| BN, ReLU, $3 \times 3$ conv, 3 Tanh |

| (b) $T_{\beta_1}$ |
| --- |
| Input: $\boldsymbol{x} \in [-1,1]^{32 \times 32 \times 3}$ |
| ResBlock down 128 |
| ResBlock down 128 |
| ResBlock down 128 |
| ResBlock 128 |
| ResBlock 128 |

| (c) $T_{\beta_2}$ |
| --- |
| Input: $\boldsymbol{x} \in \mathbb{R}^{128 \times 8 \times 8}$ |
| ReLU |
| Global sum pooling |
| $128 \to 1$ Spectral normalization |

corresponding SW and CSW variants. For the discriminator, we update $T_{\beta_1}$ and $T_{\beta_2}$ every iterations. We set the mini-batch size $m$ to 128 on CIFAR10 and CelebA, set to 32 on STL10, and set to 16 on CelebA-HQ. The learning rate of $G_\phi$, $T_{\beta_1}$, and $T_{\beta_2}$ is set to 0.0002. We use Adam [23] for optimization problems with $(\beta_1, \beta_2) = (0, 0.9)$.

**Calculation of scores:** For the FID scores and the Inception scores, we calculate them based on 50000 random samples from trained models. For the FID scores, we calculate the statistics of datasets on all training samples.

Table 9: CelebA architectures.

| (a) $G_\phi$ |
|---|
| Input: $\boldsymbol{\epsilon} \in \mathbb{R}^{128} \sim \mathcal{N}(0,1)$ |
| $128 \rightarrow 4 \times 4 \times 256$, dense linear |
| ResBlock up 256 |
| ResBlock up 256 |
| ResBlock up 256 |
| ResBlock up 256 |
| ResBlock up 256 |
| BN, ReLU, $3 \times 3$ conv, 3 Tanh |

| (b) $T_{\beta_1}$ |
|---|
| Input: $\boldsymbol{x} \in [-1,1]^{64 \times 64 \times 3}$ |
| ResBlock down 128 |
| ResBlock down 128 |
| ResBlock down 128 |
| ResBlock 128 |
| ResBlock 128 |
| ResBlock 128 |

| (c) $T_{\beta_2}$ |
|---|
| Input: $\boldsymbol{x} \in \mathbb{R}^{128 \times 8 \times 8}$ |
| ReLU |
| Global sum pooling |
| $128 \rightarrow 1$ Spectral normalization |

Table 10: STL10 architectures.

| (a) $G_\phi$ |
|---|
| Input: $\boldsymbol{\epsilon} \in \mathbb{R}^{128} \sim \mathcal{N}(0,1)$ |
| $128 \rightarrow 3 \times 3 \times 256$, dense , linear |
| ResBlock up 256 |
| ResBlock up 256 |
| ResBlock up 256 |
| ResBlock up 256 |
| ResBlock up 256 |
| BN, ReLU, $3 \times 3$ conv, 3 Tanh |

| (b) $T_{\beta_1}$ |
|---|
| Input: $\boldsymbol{x} \in [-1,1]^{96 \times 96 \times 3}$ |
| ResBlock down 128 |
| ResBlock down 128 |
| ResBlock down 128 |
| ResBlock down 128 |
| ResBlock 128 |
| ResBlock 128 |
| ResBlock 128 |

| (c) $T_{\beta_2}$ |
|---|
| Input: $\boldsymbol{x} \in \mathbb{R}^{128 \times 6 \times 6}$ |
| ReLU |
| Global sum pooling |
| $128 \rightarrow 1$ Spectral normalization |

Table 11: CelebA-HQ architectures.

| (a) $G_\phi$ |
|---|
| Input: $\boldsymbol{\epsilon} \in \mathbb{R}^{128} \sim \mathcal{N}(0,1)$ |
| $128 \rightarrow 4 \times 4 \times 256$, dense , linear |
| ResBlock up 256 |
| ResBlock up 256 |
| ResBlock up 256 |
| ResBlock up 256 |
| ResBlock up 256 |
| BN, ReLU, $3 \times 3$ conv, 3 Tanh |

| (b) $T_{\beta_1}$ |
|---|
| Input: $\boldsymbol{x} \in [-1,1]^{128 \times 128 \times 3}$ |
| ResBlock down 128 |
| ResBlock down 128 |
| ResBlock down 128 |
| ResBlock down 128 |
| ResBlock 128 |
| ResBlock 128 |
| ResBlock 128 |

| (b) $T_{\beta_2}$ |
|---|
| Input: $\boldsymbol{x} \in \mathbb{R}^{128 \times 8 \times 8}$ |
| ReLU |
| Global sum pooling |
| $128 \rightarrow 1$ Spectral normalization |