# OpenReview forum: "Revisiting Sliced Wasserstein on Images: From Vectorization to Convolution"
_NeurIPS.cc/2022/Conference — NeurIPS 2022 Accept_

### Official Review · Reviewer_s9sz · 2022-07-02

**Rating:** 7
**Confidence:** 4
**Soundness:** 3 good
**Presentation:** 4 excellent
**Contribution:** 4 excellent

**Summary:**

While the Sliced-Wasserstein distance is widely used to compare distributions on images (which can be seen as tensors), it requires to perform a vectorization step using a reshape operation which is not suited to images and loses spatial structure, while being memory inefficient. In this work, authors propose to define a Convolutional Sliced-Wasserstein on the set of probabilities over tensors using convolution operations which are better suited to images. Some properties are derived such as the sample complexity, pseudo-distance. And many experiments are performed on classical image datasets.

**Questions:**

I have first a question on the theoretical results. In Theorem 1, it is stated that it is a pseudo-distance and that $CSW_p(\mu,\nu)=0 \nLeftrightarrow \mu=\nu$. I do not find the last statement very clear. I believe that it is meant that $CSW_p(\mu,\nu)=0$ does not imply that $\mu=\nu$. I did not find in the proof of the theorem any argument supporting this statement. Is there any counter example of $\mu\neq \nu$ such that $CSW_p(\mu,\nu)=0$?

A second question is on the experiments. While the results seem quite convincing, it seems like the experiments (to get e.g. the FID and the IS) were run only once. I believe it would be more robust and convincing to take a mean over several trainings for example.






Typos:
- Line 126: $\mu\mathbb{R}^{c\times \times d}$

**Limitations:**

Yes

**Strengths And Weaknesses:**

This paper proposes to use convolutions to project images on the real line in order to use a SW distance between distributions over images. While the convolution idea is not novel since it has been widely used in neural networks for some time, the idea of using it in the context of optimal transport is new and very interesting. Moreover, the results seem quite convincing.

Strengths:
- Well written and very clear
- Using convolution to define a new SW distance.
- Application on different datasets which give good results
- Theoretical results

Weaknesses:
- Only a pseudo-distance?

---

> ### Author Response · Authors · 2022-07-29
> **Response to Reviewer s9sz**
>
> We appreciate the reviewer's time and feedback. We would like to answer the questions of the reviewer as follow:
>
> **Q16**: Pseudo distance?
>
> **A16**: Thank you for your comment. The pseudo distance means that CSW does not have the identity property. In more detail, $CSW(\mu,\nu)=0$ does not imply $\mu=\nu$. However, $\mu=\nu$ will still lead to $CSW(\mu,\nu)=0$.
>
> The CSW is a pseudo metric on the space of all distribution over tensors which means we do not assume any structure on distribution over images. In practice, many empirical investigations show that image datasets belong to some geometry group (symmetry, rotation invariant, translation invariant, and so on). Therefore, the set of distributions over images might be a subset of the set of distributions over tensors.  If the convolutional transform can hold the injectivity on the set of distributions over images, CSW can be a metric on the space of distributions over images. In our applications, we compare the value of sliced Wasserstein and convolution sliced Wasserstein on MNIST digits in Table 5 in Appendix F1, we found that the values of CSW are closed to the value of SW that can be considered as a test for our hypothesis of metricity of CSW. To our best knowledge, there is no formal definition of the space of distributions over images and its property. Therefore, we will leave this for future work.  Moreover, in deep learning applications where mini-batches are used, all optimal transport metrics will turn into a loss due to the subsampling [R.1]. Therefore, the pseudo metricity of CSW does not affect much in deep learning applications. That partially explains why CSW is still better than SW in our experiments.
>
> [R.1] Learning with minibatch Wasserstein : asymptotic and gradient properties; Fatras et al.
>
> **Q17**: the second question is on the experiments. While the results seem quite convincing, it seems like the experiments (to get e.g. the FID and the IS) were run only once. I believe it would be more robust and convincing to take a mean over several trainings for example.
>
> **A17**: Thank you for your comment. We have added the results of SW, CSW-b, CSW-s, CSW-d with (L=100, L=1000) on CIFAR10 after running 5 different times in Table 1. From the table, we still observe the same phenomenon that CSW variants are better than SW. This strengthens the claim of using convolution slicers is better than the conventional slicer in both performance and efficiency. To further verify this, we introduce max convolution sliced Wasserstein (Max-CSW) variants which are extensions of max sliced Wasserstein with convolution slicers. We also define convolution projected robust Wasserstein (CPRW) which is an extension of projected robust Wasserstein (projecting measures into $k>1$ dimensional subspace). We refer the reviewer to the definitions of Max-CSW and CPRW  in Definition 9 and Definition 11 in Appendix E, and experimental results in Table 6 in Appendix F.2. We observe that both Max-CSW variants and a CPRW variant (CPRW-stride) give better performance than the conventional approach with the vectorization step.

---

> > ### Comment · Reviewer_s9sz · 2022-08-05
> > **Response to authors**
> >
> > Thank you for your responses and explanations. I find the discussion about the pseudo-distance very interesting and I believe it would deserve to be in the paper.

---

> > > ### Author Response · Authors · 2022-08-06
> > > **Response to Reviewer s9sz**
> > >
> > > Dear Reviewer s9sz,
> > >
> > > Thank you for your positive feedback. We will revise the paper based on your comments. We would like to hear from you if you have any other concerns that want us to clarify.
> > >
> > > Best regards,

---

### Official Review · Reviewer_V4XA · 2022-07-11

**Rating:** 7
**Confidence:** 4
**Soundness:** 3 good
**Presentation:** 3 good
**Contribution:** 3 good

**Summary:**

This paper presents a new methodology for comparing two probability measures over images.
The key idea of the paper is to apply convolution operators on probability measures over images.
The proposed method, named convolution sliced Wasserstein(CSW), makes use of the spatial structure of images and needs less slicing memory.
The authors provide the metricity of CSW as well as its sample complexity, its computational complexity, and connection to sliced Wasserstein distance.
Beyond this theoretical contribution, the authors discuss numerical considerations and perform a thorough real data analysis, showing the favorable performance of their method.

**Questions:**

**Major Comments**

* I am a little confused about the definition of the convolution-base slicer and its invariants.
In definition 3, when $d$ is even, the last kernel $K^{(N)}$ has dimension $1\times a \times a$.
However, $a=\frac{d}{2^{N-1}}$ and by definition $a$ is an integer.
Is there any guarantee about this?
For example, what is the definition of these kernels when $d=18$?
In real data, the dimension $d=32,64,96,128$,
whose prime factor decomposition has at most one number which is not 2.
But $a=\frac{d}{2^{N-1}}$ won't be an integer when $d=18=2\times 3\times 3$.
This question also lies in the definition of other variants of convolution sliced-Wasserstein distances.

* The authors fix the stride size $s=2$ in definition of convolution-stride slicer.
There may be some trade-off between time consumption and estimation efficiency for different stride sizes $s$.
Some simple analysis may help us recognize it intuitively.
I am wondering how does the stride size affect the performance of the proposed method? Some discussion and empirical results are needed.

* This paper mainly considers probability measures over images. It would be convincing if the authors can provide other tasks in real data applications.



**Limitations:**



**Strengths And Weaknesses:**

The idea of applying the convolution operators on probability measures over images is a very natural idea.
The proposed method is novel and simple to describe.
Also, the authors show the efficiency numerically.

---

> ### Author Response · Authors · 2022-07-29
> **Response to Reviewer V4XA**
>
> We appreciate the reviewer's time and feedback. We would like to answer the questions of the reviewer as follow:
>
> **Q13**: I am a little confused about the definition of the convolution-base slicer and its invariants… what is the definition of these kernels when d=18?
>
> **A13**:  Thank you for your comment. The definition of slicers is actually a CNN that maps the images to a one-dimensional scala, hence, it is flexible to choose the size of the kernels in each layer. In our three proposed slicers, we aim to reduce the dimension by half after each layer. When the dimension of the feature map is not even, we will use the kernel size that is equal to the size of the feature map to map the dimension to 1. For example. $d=18=2 \times 9$. Therefore, our proposed convolution slicers will have 2 layers.
>
> For Convolution-base Slicer, the first kernel size is $10\times 10$  and the second kernel size is $9\times9$.
> For Convolution-stride slicer, the first kernel size is $2\times 2$  and the second kernel size is $9\times 9$.
> For Convolution-dilation slicer, the first kernel size is $2\times 2$  and the second kernel size is $9 \times 9$.
>
> We have revised the definition of the number of layers $N$, and $a$ in the revision as follows. Given the dimension $d$, $N$ is the biggest integer that satisfies $d=2^{N-1}\times a$, where $a$ is also an integer. Hence, $N$ is the number of layers and $a$ is the size of the last kernel. The size of the intermediate kernel is unchanged. We have updated the definition to the revision in blue color, please check it for a more formal definition.
>
> **Q14**: The authors fix the stride size s=2 in the definition of convolution-stride slicer. There may be some trade-off between time consumption and estimation efficiency for different stride sizes s. Some simple analysis may help us recognize it intuitively. I am wondering how does the stride size affect the performance of the proposed method? Some discussion and empirical results are needed.
>
> **A14**: Thank you for your comment. As mentioned in the A13, the definition of slicers is flexible as designing a CNN. In practice, the architecture depends on a lot of factors including tasks, datasets, and so on. In the paper, we propose three simple variants which might be not the best choice. We only aim to show that using convolution can easily beat conventional vectorization slicing. Using stride can get rid of the dependency of the convolution kernel size to the dimension. A large stride can help to reduce the dimension faster, however, it might lose some local information. To our best knowledge, the effect of stride size has not been established in the literature yet. Therefore, we leave this question to future work.  Due to the limitation of time and hardware, we have not been able to run experiments on generative modeling with different stride configurations. We will try to report the result to the discussion when it is available.
>
> **Q15**: This paper mainly considers probability measures over images. It would be convincing if the authors can provide other tasks in real data applications.
>
> **A15**: Similar to SW, CSW can be applied to other applications such as autoencoders [R.3], domain adaptation [R.4], and so on. When dealing with images, CSW is a better choice than SW. When dealing with a general type of data, it is not guaranteed that the performance of CSW is better than SW. However, CSW will always be more efficient in computation and memory. We would like to recall that CSW is the first step to incorporating geometry and inductive bias into sliced Wasserstein distance. It will be a very interesting direction in designing variants of SW with geometry-oriented projection in other types of data such as text, time series, and graphs.  In these scenarios, special architectures can be used such as Transformer, recurrent neural networks, graph neural networks, and so on. These distances can be used in various applications that deal with probability measures such as generative modeling, clustering, domain adaptation, adversarial attacks, the point could reconstruction, and so on. However, there will be also challenges such as designing the slicing distribution and controlling the projection complexity (the number of parameters of slicers). We leave these investigations to future works.
>
> [R.3] Sliced Wasserstein Auto-Encoders; Kolouri  et al
>
> [R.4] Sliced Wasserstein Discrepancy for Unsupervised Domain Adaptation; Lee et al

---

> > ### Comment · Reviewer_V4XA · 2022-08-09
> > **response to rebuttal**
> >
> > Thank you for your detailed explanations. I have no more questions and would tend to keep the score.

---

> ### Author Response · Authors · 2022-08-06
> **Look forward to your feedback.**
>
> Dear Reviewer V4XA,
>
> We have addressed your concerns in our responses. We would like to hear your feedback. Please feel free to raise questions if you have other concerns.
>
> Best regards,
>
> Authors

---

### Official Review · Reviewer_nLU5 · 2022-07-11

**Rating:** 4
**Confidence:** 2
**Soundness:** 3 good
**Presentation:** 3 good
**Contribution:** 2 fair

**Summary:**

This paper proposed convolutional sliced Wasserstein distance and compared it with the conventional sliced Wasserstein distance. In addition, the authors also introduced a convolution base slicer, a convolution stride slicer, and its variant with dilation. The paper provided the details about how the convolutional sliced Wasserstein is calculated, and analyzed several properties. The experiment results on the multiple datasets demonstrate that for the generative tasks, the proposed CSW achieved better performance than SW while using similar computational resources.

**Questions:**

See the weakness section.

Overall this paper proposed an interesting idea to better compute the Wasserstein distance between two image sets. However I lack the related background knowledge and some experiment details are missing, I cannot judge the merit of this paper.

**Limitations:**

The authors did not provide the discussions or analysis about the limitations and potential negative societal impact (if I did not miss anything). Overall I think it is fine as the paper focuses more on the fundamental SW distance computation. See the weakness section for potential technical limitations, for example, the choice of kernels.

**Strengths And Weaknesses:**

The paper is well written and provides a lot of details about SW, which is very helpful for readers like me with little SW background.
The idea is well motivated and the math seems convincing to me.

I must apologize I don't have enough expertise to judge the merit of this paper. Below are some of my questions and concerns.
1. In L201, the authors did notice that convolution is a linear operation. Thus consecutive convolution operations can be replaced with a single convolution. Except for the non-linear convolution slicer, what is the purpose of using multiple linear convolution operations? Moreover, convolution is often implemented as matrix multiplication, which also flattens the image and converts the kernel into a Toeplitz matrix. So the proposed convolutional SW is similar to SW with a special R in Eq. 1?

2. I wonder how the kernel is determined? From L228-230, it seems the kernel is uniformly sampled from the set $K^{(l)}$. Do we sample once for all the images (that is, using the same set of kernels for all the images) or we repetitively sample kernels for each image?

3. Some training details are missing. What is the loss function? Based on the main script, it seems the network does not need a discriminator as CSW directly computes the distance between the generated images and real images and the network is trained to minimize the distance. However, in the supplementary, discriminators are still used (L732-739). If discriminators are used, I am a little confused as SW GAN [13] used discriminators to map images into 1D vector and compute SW, which is similar to nonlinear CSW introduced in this submission, except that kernels in [13] are learnable while the kernels in this submission is uniformly sampled.

4. The authors might miss one reference about introducing convolution in Wasserstein to reduce the computational cost:
Justin Solomon, Fernando de Goes, Gabriel Peyré, Marco Cuturi, Adrian Butscher, Andy Nguyen, Tao Du, Leonidas Guibas, Convolutional Wasserstein Distances: Efficient Optimal Transportation on Geometric Domains, Proc. SIGGRAPH 2015.

Minor: L126 $R^{c\times \times d}$ should be $R^{c\times d\times d}$

---

> ### Author Response · Authors · 2022-07-29
> **Response to Reviewer nLU5 - Part 1**
>
> Thank you for your time and feedback. We would like to answer the questions of the reviewer below. We also would like to mention that we have applied convolution slicing to define new discrepancies which are Max convolution sliced Wasserstein (Max-CSW) and Convolution projected robust Wasserstein (CPRW). They are based on max sliced Wasserstein (Max-SW) and projected robust Wasserstein (PRW) which use the conventional vectorization approach. The details of definitions are given in Definition 9 and Definition 11 in Appendix E. We conducted experiments to compare them with Max-SW and PRW in Table 6 in Appendix F.2 and observed a better result for our proposed discrepancies.
>
> **Q7**: In L201, the authors did notice that convolution is a linear operation. Thus consecutive convolution operations can be replaced with a single convolution. Except for the non-linear convolution slicer, what is the purpose of using multiple linear convolution operations?
>
> **A7**: Thank you for a detailed and interesting question. Consecutive convolution operations can be replaced with a single convolution; however, the distribution of that “single convolution”  can be complicated. To ease the ensuing discussion, we refer to the “single convolution” as the aggregated projecting direction variable. In contrast to SW which uses the uniform distribution on the aggregated projecting direction variable which is shown to be redundant and not efficient in high-dimension [R.5,R.7], CSW uses a more complicated and geometry-oriented distribution. Therefore, CSW can be seen as a special case of distributional sliced Wasserstein (DSW) [R.8] where the distribution over the aggregated projecting direction is not uniform and is not supported on the unit-hypersphere. Also, the distribution over the aggregated projecting direction is regularized implicitly by architectures while DSW uses a loss-regularizer. We would like to recall that the other benefit of using convolution is saving computation and memory.
>
> [R.5] Generalized Sliced Wasserstein Distances; Kolouri  et al
>
> [R.7] Max-Sliced Wasserstein Distance and its use for GANs; Deshpande et al
>
> [R.8] Distributional sliced Wasserstein and Applications to Generative modeling; Nguyen et al
>
> **Q8**: Moreover, convolution is often implemented as matrix multiplication, which also flattens the image and converts the kernel into a Toeplitz matrix. So the proposed convolutional SW is similar to SW with a special R in Eq. 1?
>
> **A8**: Thanks for your question. As in **A7**, CSW can be seen as a special case of DSW which is SW with a special R in Eq. 1. Convolution is often implemented as matrix multiplication; however, the Toeplitz matrix of convolution is very sparse which improves the computational time and memory of matrix multiplication. Moreover, in some special computational hardware that supports computing convolution directly without converting to the Toeplitz matrix, the computational time of CSW will be improved further.
>
> **Q9**: I wonder how the kernel is determined? From L228-230, it seems the kernel is uniformly sampled from the set K(l) Do we sample once for all the images (that is, using the same set of kernels for all the images), or do we repetitively sample kernels for each image?
>
> **A9**: For each estimation of CSW, the kernel is sampled once. However, in applications, where the computation of CSW is repeated on multiple mini-batches, the kernel is sampled repeatedly for each mini-batch. We would like to recall that we use group convolution to make sure that $(K_i^{(1)},...,K_i^{(N)})_{i=1}^L$ are independent for the Monte Carlo estimation.

---

> ### Author Response · Authors · 2022-07-29
> **Response to Reviewer nLU5 - Part 2**
>
> **Q10**: Some training details are missing. What is the loss function? Based on the main script, it seems the network does not need a discriminator as CSW directly computes the distance between the generated images and real images, and the network is trained to minimize the distance. However, in the supplementary, discriminators are still used (L732-739). If discriminators are used, I am a little confused as SW GAN [13] used discriminators to map images into 1D vectors and compute SW, which is similar to nonlinear CSW introduced in this submission, except that kernels in [13] are learnable while the kernels in this submission are uniformly sampled.
>
> **A10**: In the experiments, we also use the discriminator which is trained by the loss given in Appendix F.2. The generator is trained by using SW and CSW on the feature space of the discriminator. We agree that this is similar to the idea of nonlinear CSW; however, in this case, the space of projections is over-parametrized and only one projection (slice) is selected by maximizing a training loss. The minimax training can be seen as a Max-sliced variant of Non-linear CSW with a more rich architecture for the slicer. we would like to mention that almost all deep learning applications uses sliced Wasserstein on vectorized features of a non-linear convolutional neural network which can be considered a more complex form of non-linear CSW (e.g., deep generative model [R.2],[R.3], deep domain adaptation [R.4]).
>
> The reason that we need non-linearity is that using linear projection and $\mathbb{L}_2$ ground metric cannot produce good images since the geodesic metric between images is often non-trivial [R.9]. Therefore, in practice, practitioners need to use the discriminator as the non-linear projection and the ground metric learning [R.2,R.10]. As mentioned above, the architecture of the discriminator is very complicated, hence, it is hard to formulate practical training in a rigorous formulation based on the current literature. The definitions of CSW and Non-linear CSW is the first building block to understanding the training process. Moreover, since the discriminator also uses convolution, it reinforces the claim that we cannot use directly  SW with vectorization in applications on images. In the paper, we choose the setting from [R.2] which can produce good generated images despite being not fair for CSW. However, CSW still performs better than SW which indicates the importance of convolution slicing.
>
> [R.2] Generative Modeling using the Sliced Wasserstein Distance; Deshpande et al
>
> [R.3] Sliced Wasserstein Auto-Encoders; Kolouri et al
>
> [R.4] Sliced Wasserstein Discrepancy for Unsupervised Domain Adaptation; Lee et al
>
> [R.9] Wasserstein GANs Work Because They Fail (to Approximate the Wasserstein Distance); Stanczuk et al
>
> [R.10] Learning Generative Models with Sinkhorn Divergences; Genevay et al
>
> **Q11**: The authors might miss one reference about introducing convolution in Wasserstein to reduce the computational cost: Justin Solomon, Fernando de Goes, Gabriel Peyré, Marco Cuturi, Adrian Butscher, Andy Nguyen, Tao Du, Leonidas Guibas, Convolutional Wasserstein Distances: Efficient Optimal Transportation on Geometric Domains, Proc. SIGGRAPH 2015.
>
> **A11**: Thank you for the reference, we have cited it in related work in Appendix A in the revision. In the mentioned paper, the convolution is used to find the ground metric of optimal transport based on solving the heat equation. The resulted distance has the complexity of $\mathcal{O}(n^2)$ where n is the number of supports. Our paper uses a convolution operator in a different way which is a slicing method (or a dimension reduction method) to map original measures to multiple one-dimensional measures. Since the computation of Wasserstein distance on one-dimension has closed form $\mathcal{O}(n\log n)$, our resulted distance costs only $\mathcal{O}(Ln\log n)$ where $L$ is the number of projections.
>
> **Q12**: limitations and potential negative societal impact?
>
> **A12**: We leave the discussion on negative societal impact in Appendix A. Overall, our proposed CSW is a general tool, hence, it could be used in applications that do not have a good purpose. For the limitations of CSW, we mention that CSW is not a metric on the space of all distributions over tensors and it might only be meaningful when used on images. The reason is that the set of distributions over images might be a subset of the set of distributions over tensors which are invariant to rotation, translation, and so on.  For general distributions, the superior performance of CSW compared to SW is not guaranteed and CSW will only be better than SW in terms of computation and memory. We refer the reviewer to **Q2** of Reviewer GwEg for a detailed discussion. We have also added a paragraph to discuss the limitations in Appendix A in the revision.

---

> ### Author Response · Authors · 2022-08-05
> **Looking forward to your feedback**
>
> Dear Reviewer nLU5,
>
> We would be grateful if you give us your feedback about our response. We believe that we have addressed all your concerns in your reviews. Please feel free to raise questions if you have other concerns.
>
> Best regards,
>
> Authors,

---

> ### Author Response · Authors · 2022-08-08
> **Looking forward to your feedback**
>
> Reviewer nLU5,
>
> We have addressed your concerns in our responses. Given that the discussion deadline is approaching, we would like to hear your feedback. Please feel free to raise questions if you have other concerns.
>
> To summarize, we have addressed the connection between CSW to DSW. Moreover, we discuss carefully the benefit of using the convolution directly on **tensors** compared to using Toeplitz matrix multiplication on **vectors**. Namely, the direct computation of convolution has lower memory complexity and time complexity than using Toeplitz matrix multiplication. Also, the convolution can be implemented on hardware to speed up the computation [R.1,R.2,R.3]. In addition, we have discussed in detail the projection sampling step and the training process of generative modeling.
>
> [R.1] Accelerating Deep Convolutional Neural Networks Using Specialized Hardware, Ovtcharov et al.
>
> [R.2] Hardware Accelerated Convolutional Neural Networks for Synthetic Vision Systems, Farabet et al.
>
> [R.3] A Unified Hardware Architecture for Convolutions and Deconvolutions in CNN, Bai et al
>
> Best regards,
>
> Authors

---

> > ### Comment · Reviewer_nLU5 · 2022-08-09
> > **Efficiency and main concern**
> >
> > Dear authors,
> >
> > I really appreciate the detailed feedback and it addressed some of my concerns. I apologize that I thought the discussion deadline is the 19th as that shown on my task panel so I prioritized other tasks.
> >
> > 1. Efficiency
> > How to implement convolution is beyond the scope of this paper. For example, TensorFlow implements conv2d as GEMM (though not Toeplitz):
> > https://github.com/tensorflow/tensorflow/blob/v2.9.1/tensorflow/python/ops/nn_ops.py#L2238-L2249
> > (this is a correlation, not convolution, more specifically)
> >
> > In certain cases, FFT/IFFT implementation might be more efficient (I think this might be more widely used in the communication system like MIMO), for example:
> > https://ieeexplore.ieee.org/document/8653873
> >
> > For [R.1] Accelerating Deep Convolutional Neural Networks Using Specialized Hardware, Ovtcharov et al., if you look at table 1, FPGA only won at power consumption. (you might argue the cuDNN contributed most to the efficiency and I agreed with you as I thought cuDNN implements convolution as 4 nested loops. Sorry I could not find its code to back this.)
> >
> > Anyway, my point in my initial review is you can use a single convolution to replace your consecutive linear convolution. If I understood correctly from the rebuttal, you argued that consecutive convolution with the proposed kernel sampling scheme has some beneficial distribution properties. This seems a valid argument but I cannot judge the correctness of this claim either due to the lack of expertise (I don't think I can gain all the necessary knowledge in one or two weeks either).
> >
> > 2. CSW in discriminators.
> > I may share a common concern with the reviewer GwEg:
> >
> > based on the loss function in appendix F.2, you still need a couple of learnable convolution steps ($T_{\beta_1}$) to obtain the image feature as the **tensor** format (thank authors for pointing this loss function as I missed it in my initial review) to compute CSW. What is the difference between these convolution steps and the convolution in CSW?
> >
> > Will the result be better if you use more of these learnable convolution steps (i.e. deeper networks)?
> >
> > If so, we can use a deep enough network to directly map the images into 1d **vectors** and apply SW and hopefully get better results (in this case, the hypothesis is the learned convolution kernel weights are better than those randomly sampled in CSW).
> >
> > If not, then why not directly apply CSW on the generated images (in this case, the hypothesis is the randomly sampled convolution kernel weights are better than the learned)?
> >
> > Or is there a "sweet point" that a proper combination of the learned convolution and random sample convolution in CSW applying to the images yields the best results?

---

> > > ### Author Response · Authors · 2022-08-09
> > > **Response to Reviewer nLU5**
> > >
> > > Dear Reviewer nLU5,
> > >
> > > Thank you for the detailed comments about the practical implementation of convolution. The point we want to make is that defining sliced Wasserstein variants on tensors can allow us to have more efficient ways to project distributions into one-dimensional space. Convolution is one example where doing vectorization leads to a worse computational complexity and memory complexity with Toeplitz multiplication. We would like to answer your questions as follow:
> > >
> > > **Q1** What is the difference between these convolution steps and the convolution in CSW?
> > >
> > > **A1**  First, we would like to recall that we follow the setup in [R.2] which gives the most realistic generative images for sliced Wasserstein generative models. The difference between the convolution steps and the convolution in CSW is that the weights of convolution steps are learned by a GAN loss while the weights of the convolution in CSW are sampled randomly. Moreover, the weights of convolution steps do not have any constraints while the weights of the convolution in CSW have the sum of square equals 1. The meaning of the convolution steps is to map original distributions into a feature space (high-dimension) where pushforward distributions can be linearly separated by sliced Wasserstein. In contrast, the meaning of the convolution in CSW is to map feature distributions to one dimension for a closed form of Wasserstein distance for comparing two distributions.
> > >
> > > [R.2] Generative Modeling using the Sliced Wasserstein Distance; Deshpande et al
> > >
> > > **Q2** Will the result be better if you use more of these learnable convolution steps (i.e. deeper networks)?
> > >
> > > **A2** To our best knowledge, the answer to this question has not been investigated yet. We follow the standard setup in [R.2] to compare CSW to SW despite the fact that the setting is not fair for CSW since using feature extraction convolution steps can be considered a more complicated version of CSW. We agree that the practical algorithm has not been able to formulate in a formal mathematical formulation yet. However, CSW is the first building block to explain it.
> > >
> > >
> > > **Q3**  ... using a deep enough network to directly map the images into 1d vectors and apply SW and hopefully get better results.
> > >
> > > **A3** Using a deep neural network naively could lead to a high projection complexity (the space of all weights of the neural network), hence, it is impossible to do sampling and apply Monte Carlo estimation. Moreover, without constraints on the weights of neural networks, statistical estimation rate (sample complexity) cannot be derived. In CSW, we can derive it in Proposition 2 by setting a norm constraint on the weights of the convolution kernel. To our best knowledge, there is no work that has done this before.
> > >
> > > **Q4** why not directly apply CSW on the generated images (in this case, the hypothesis is the randomly sampled convolution kernel weights are better than the learned)?
> > >
> > > **A4** Sliced Wasserstein variants require a ground metric in the projected space which is normally $\mathbb{L}_2$ distance. However, this choice is not good for images since the geodesic distance between images is much more complicated and is normally unknown [R.1*]. Therefore, in deep generative modeling, the geodesic distance is implicitly learned by using a discriminator. The non-linearity inside the discriminator and its deep architecture can form a complex type of metric learning.  Without using a discriminator, trained generative models will favor the $\mathbb{L}_2$ distance between images that create non-realistic images [R.1*].
> > >
> > > [R.1*] Wasserstein GANs Work Because They Fail (to Approximate the Wasserstein Distance), Stanczuk et al.
> > >
> > > **A5** is there a "sweet point" that a proper combination of the learned convolution and random sample convolution in CSW applying to the images yields the best results?
> > >
> > > **Q5** This is an interesting question. However, we do not have the answer to this at the moment. We follow the standard setup in [R.2] to compare CSW with SW. To our best knowledge, this is the only way to train generative models that produce realistic images. A lot of works also follow this setup [R.2*,R.3*,R.4*]
> > >
> > > [R.2*] Distributional sliced Wasserstein and Applications to Generative modeling; Nguyen et al
> > > [R.3*] Augmented Sliced Wasserstein Distances, Chen et al
> > > [R.4*] Fast Approximation of the Sliced-Wasserstein Distance Using Concentration of Random Projections, Nadjahi et al
> > >
> > > Overall, we would like to make the point that **sliced Wasserstein has not been ever defined on **tensor** space**, hence, practical training approaches cannot be explained yet. We make the first step to explain why the practical training approach works. Again, we would like to mention that using convolution steps for feature extracting is not a fair setup for CSW, however, it is still better than SW. This reinforces the need of defining sliced Wasserstein variants on tensors without doing vectorization.
> > >
> > > Best regards,
> > >
> > > Authors

---

> > > > ### Comment · Reviewer_nLU5 · 2022-08-09
> > > > **SW on tensor space**
> > > >
> > > > Thanks for the prompt response.
> > > >
> > > > I understood the authors' point is to emphasize that this paper is the first to define SW on the tensor space using convolution.
> > > >
> > > > The reason why I proposed the previous hypotheses is: if in practice, using a deep network to (non-linearly) map the tensor into a vector then applying conventional SW is good enough, defining SW on tensor space might be a fake problem.
> > > >
> > > > As a researcher with an application-oriented mindset, it really took me some time to understand the theoretical merit of this paper, though my understanding is still very limited.
> > > >
> > > > Anyway I really appreciate all the discussions.

---

> > > > > ### Author Response · Authors · 2022-08-09
> > > > > **Response to Reviewer nLU5,**
> > > > >
> > > > > Dear Reviewer nLU5,
> > > > >
> > > > > Thank you for your quick response,
> > > > >
> > > > > Our point is that the composition of the **convolution** neural network (non-linearly)  that maps the tensor into a vector and then uses SW is a generalized version of convolution sliced Wasserstein. Here,  the projection space is over-parametrized and only one instance of that space is used (found by an optimization problem). The thing is that using the **convolution** neural network accepts the input as a distribution over **tensor** implicitly.
> > > > >
> > > > > Best regards,
> > > > >
> > > > > Authors,

---

> > > ### Author Response · Authors · 2022-08-09
> > > **Please feel free to ask other questions if you still have other concerns**
> > >
> > > Dear Reviewer nLU5,
> > >
> > > We appreciate your time and feedback. We would like to thank you for giving us a chance to clarify the contribution of our work, implementation detail, related works, and discussion. We are looking for work your feedback.
> > >
> > > Best regards,
> > >
> > > Authors,

---

### Official Review · Reviewer_GwEg · 2022-07-15

**Rating:** 3
**Confidence:** 3
**Soundness:** 2 fair
**Presentation:** 2 fair
**Contribution:** 1 poor

**Summary:**

This work attempted to extend the sliced Wasserstein distance to be the convolution sliced Wasserstein distance by replacing vectors with tensors and Radon transforms with convolution operators, respectively. Then, this work compared its proposed methods with different convolutions (usual convolution, convolution with stride, convolution with dilation) with the original sliced Wasserstein on image generation task with various datasets (CIFAR10, CelebA, STL10, CelebA-HQ), claiming to show favorable results. Its theorem showed that this new convolution sliced Wasserstein distance is a pseudo-metric, it is always less than or equal to the max sliced Wasserstein distance, and its expectation is upper bounded with a specific value.

**Questions:**

- Feel free to argue back to the weakness points above.
- This work argued the needs for using tensors instead of vectors in the sliced Wasserstein distance. While it might be making sense, it was not easy for me to be convinced about these arguments. Conventional image processing works have widely used vector representations in their models and tensors (i.e., reshapes) in their implementations. While this manuscript mentioned "a probability measure over images should be defined over the space of tensors instead of images" and "this extra step does not keep the spatial structures of the supports, which are crucial information of images", I need more explanation on why these sentences are justified. Even though images are represented as vectors, spatial structures can be considered in the operations on them. In that sense, these arguments seem relatively weak.

**Limitations:**

Even though the authors indicated that the limitations and potential negative societal impact of their work, I was not able to find them.

**Strengths And Weaknesses:**

- Even though this manuscript claimed that "We derive convolution sliced Wasserstein (CSW) and its variants" in the abstract, CSW was not derived, but was defined in this work (see Definition 5). While this work seems to assume that the distance between "images" are compared with the Wasserstein distance, there are a number of cases where "vectorized feature tensors" of two images are compared with the (sliced) Wasserstein distance (e.g., see [35]) and in that case, this work can be seen as a simplified version of these works. In that sense, it can be seen that there is no novelty in the definitions of this manuscript.
- While this defined CSW was shown to be a pseudo-metric in Theorem 1, it may be a serious weakness as a loss to be used in the optimization. Will the different image with the distance 0 be good to represent Unfortunately, there is no discussion on this issue. Non-linear extension of CSW was mentioned in lines 328-332, it does not seem to be meaningful as a metric considering that linear version of it is a pseudo-metric.
-  While its experiments were showing results on image generation task for various datasets, it was only compared with a conventional SW method, not a recent max SW or generalized sliced Wasserstein [23]. Moreover, there were many different convolutions defined in the main text and compared in the experiments, the results seems to suggest no clear winner among these convolutions. Thus, the experiments in its current form seem weak. Comparing with recent Wasserstein variants and investigating proper (or optimal) convolution operators could improve this manuscript.

---

> ### Author Response · Authors · 2022-07-29
> **Response to Reviewer GwEg - Part 1**
>
> We would like to say thank you for your time and your reviews.
>
> **Q1**:  ..., CSW was not derived, but was defined in this work (see Definition 5). While this work seems to assume that the distance between "images" is compared with the Wasserstein distance, there are a number of cases where "vectorized feature tensors" of two images are compared with the (sliced) Wasserstein distance (e.g., see [35]) and in that case, this work can be seen as a simplified version of these works. In that sense, it can be seen that there is no novelty in the definitions of this manuscript.
>
> **A1**: Thanks for your comments. We agree that there are previous works that use (sliced) Wasserstein distance on images. However, all those works need to vectorize images or feature tensors of images into vectors since the conventional definition of (sliced) Wasserstein is defined between distributions on vector space. Due to the projection step in sliced Wasserstein distance, there are two limitations of this approach when we consider sliced Wasserstein distance between images. Firstly, the vectorization steps might destroy the geometry structure of images which are shown to be important in image processing and computer vision literature. Secondly, the slicing directions of sliced Wasserstein must be in the same dimension as the images which is not efficient in terms of memory and computation.
>
> Our main contribution is to get rid of the vectorization step. To do that, instead of dealing with distributions on vector space, we first need to define the discrepancy between distributions over tensor space. We then need to design the projection step that maps directly a tensor to a one-dimensional scalar. Since the convolution operator has been successfully used for images, we choose to map images via multiple convolution operators with uniform random kernels. This design helps to preserve the geometry of images and also helps to save computation and memory as in Proposition 3 in Appendix B.

---

> > ### Comment · Reviewer_GwEg · 2022-08-08
> > **Comments on "Response to Reviewer GwEg - Part 1"**
> >
> > I would like to thank the authors for their long, detailed responses. Sorry for late reply, but it took me a while to read and contemplate the manuscript and the responses. While this response clearly revealed that vectorization - projection was replaced with convolutions, unfortunately, I found great similarities between this proposed work and the work of [23], "Generalized sliced Wasserstein distances". The work of [23] proposed generalized sliced Wasserstein distances with a neural-network based projection scheme including convolutional networks with leaky ReLU activations. In other words, the proposed method in this manuscript can be seen as a special case of the work of [23] with a single layer convolutional neural network and linear activations (which is a case of leaky ReLU). The work of [23] also showed that this neural-network based projection is a pseudo-metric, which is consistent with the Theorem 1 in this manuscript. Thus, I still can not be convinced that this work contains novel contributions. Could you response to this comment if possible?

---

> > > ### Author Response · Authors · 2022-08-08
> > > **Response to Reviewer GwEg**
> > >
> > > Thank you for your response. We would like to clarify our contribution of CSW compared to the neural network version of generalized sliced Wasserstein [23].
> > >
> > > [23] "Generalized sliced Wasserstein distances"
> > >
> > > 1. The authors in [23] did not define the space of projection rigorously for the neural network which makes the study of sample complexity impossible. In contrast, we define rigorously the projection space of CSW which is the set of all convolution kernels that has the norm 2 square equals 1. This rigorous definition helps us to derive the sample complexity of CSW in Proposition 2.
> > >
> > > 2. The projection complexity of a general neural network-based projection is high. For example, the authors in [23] propose to use a multi-layer perception (MLPs) with Leaky-ReLu activation that leads to a very large number of parameters for the space projection which is the space of all possible values of the weights of the MLPs. Therefore, the computational complexity and projection complexity are huge compared to the conventional sliced Wasserstein. In contrast, our CSW has a lower computational complexity and projection complexity than the conventional SW due to the lightweight convolution kernel. In [23], the authors must use the max slice variant (Max-GSW-NN) for saving the memory, however, the authors conducted experiments only on gradient flow application which is not very large scale.  Moreover, the authors in [23] did not publish the code for that application https://github.com/kimiandj/gsw. In the deep generative modeling setting, the performance in terms of generative quality of Max-GSW-NN is only slightly better than SW while the computational time is much slower [R.8]. In contrast, CSW is better than SW while being faster.
> > >
> > > [R.8] Distributional sliced Wasserstein and Applications to Generative modeling; Nguyen et al
> > >
> > > 3. Again, the most important contribution is that GSW-NN is still defined on **vector** space. It is the reason why the authors in [23] propose to use MLPs instead of convolution neural networks.  In contrast, we are the first people that define a sliced Wasserstein variant between distributions over **tensors** or images. We would like to recall that one of our contributions is changing the way of defining and interpreting sliced Wasserstein variants on different types of data.  It is not just about choosing the architecture of neural work.  The most crucial message that we want to bring in is incorporating geometry into the design of slicing operators for different types of data.
> > >
> > > 4. The generalized sliced Wasserstein defines with a general slicing function $f_\theta$. Hence, everything slicing function can be considered a special case of $f_\theta$. However, designing an efficient parameterization of $f_\theta$  is a challenge that has not been solved and that challenge has been prevented the applications of GSW-NN in practice. In the paper, we make the first step to shedding light on the "black-box" $f_\theta$ and make it possible to be used in large-scale applications.
> > >
> > >
> > > Overall, our contribution is new and novel in the sliced Wasserstein literature. We appreciate the reviewer for giving us the opportunity to clarify our work. Please feel free to raise questions if you still have other concerns.
> > >
> > > Best regards,
> > >
> > > Authors,

---

> > > > ### Comment · Reviewer_GwEg · 2022-08-08
> > > > **Need your response for the following logic**
> > > >
> > > > Thanks for your quick replies! Could you add some comments / responses on the following logic "the proposed method in this manuscript can be seen as a special case of the work of [23] with a single layer convolutional neural network and linear activations (which is a case of leaky ReLU)"? Your response will be helpful for me to evaluate the theoretical novelty of the proposed method.

---

> > > > > ### Author Response · Authors · 2022-08-08
> > > > > **Response to the reviewer**
> > > > >
> > > > > Thank you for your quick response,
> > > > >
> > > > > Concretely, the work of [23] cannot use the 2D convolution operator since it is defined on vectors. Moreover, we would like to recall that our work uses multiple layers of convolution instead of only one layer.
> > > > >
> > > > > We will add this discussion to the revision of our paper.
> > > > >
> > > > > Best regards,
> > > > >
> > > > > Authors,

---

> > > > > > ### Comment · Reviewer_GwEg · 2022-08-08
> > > > > > **Still have the same question...**
> > > > > >
> > > > > > Thanks for the reminder on multi-layer convolutions of the proposed method. However, my original question has not been lifted yet.
> > > > > > As a simple example, a matrix multiplied by a vector can easily implement a 2D convolution via "conventional lexicographic ordering" of the vector (so the matrix will be sparse). In that sense, there is no theoretical issue in using performing 2D convolutions on vectors (see the following example: https://stackoverflow.com/questions/16798888/2-d-convolution-as-a-matrix-matrix-multiplication). Thus, unlike the authors' argument above, I can not see any problem for the work of [23] to be applicable to 2D convolutions as claimed in that work. Could you response to this?

---

> > > > > > > ### Author Response · Authors · 2022-08-08
> > > > > > > **Response to the reviewer**
> > > > > > >
> > > > > > > Thank you for your response,
> > > > > > >
> > > > > > > Theoretically, 2D convolution can be converted to matrix multiplication. However, that will create a lot of redundantly zero multiplication which worsens the computation complexity and the memory complexity. This is also an issue of the vectorization step due to defining the sliced Wasserstein variant on **vectors**.  We would like to recall that convolution can be implemented directly on the hardware [R.1,R.2,R.3].
> > > > > > >
> > > > > > > Again, we would like to recall that the work [23] did not define the space of projection rigorously. Hence, the variant of using neural networks in [23] is not understandable in terms of sample complexity which controls the statistical estimation rate. Therefore, it is impossible to verify whether the neural network variant in [23] can avoid the curse of dimensionality or not. In contrast, CSW can be proved to be able to escape the curse of dimensionality like in Proposition 2 in the paper. Overall, we believe that the view of changing the definition from the conventional definitions from **vector** space to **tensor** space is new and subtle. Please feel free to ask more questions. We appreciate the time and effort of the reviewer in clarifying our paper.
> > > > > > >
> > > > > > > [R.1] Accelerating Deep Convolutional Neural Networks Using Specialized Hardware, Ovtcharov et al.
> > > > > > >
> > > > > > > [R.2] Hardware Accelerated Convolutional Neural Networks for Synthetic Vision Systems, Farabet et al.
> > > > > > >
> > > > > > > [R.3] A Unified Hardware Architecture for Convolutions and Deconvolutions in CNN, Bai et al
> > > > > > >
> > > > > > > Best regards,
> > > > > > >
> > > > > > > Authors,

---

> > > > > > ### Author Response · Authors · 2022-08-08
> > > > > > **Example of comparison convolution to the corresponding matrix multiplication**
> > > > > >
> > > > > > Dear Reviewer GwEg, we would like to give an example for demonstrating the benefit of using convolution directly without doing vectorization.
> > > > > >
> > > > > > For example, a tensor of size $c \times d \times d$ ($d$ is even) can be mapped to a tensor of size $1 \times d/2 \times d/2$ with the kernel of size $c\times 2 \times 2$ and stride $2$ (invariance to the dimension $d$) and with time complexity $\mathcal{O}\left(cd^2 \right)$. When using Toeplitz matrix multiplication, the size of the  Toeplitz matrix is $cd^2 \times d^2/4$ which leads to the time complexity of $\mathcal{O}\left(\frac{1}{4}c d^4 \right)$.
> > > > > >
> > > > > > Best regards,

---

> > > > > ### Author Response · Authors · 2022-08-08
> > > > > **The benefit of using multiple layers of convolution**
> > > > >
> > > > >
> > > > > As in discussion with Reviewer nLU5, consecutive convolution operations can be replaced with a single convolution; however, the distribution of that “single convolution” can be complicated. To ease the ensuing discussion, we refer to the “single convolution” as the aggregated projecting direction variable. CSW can be seen as a special case of distributional sliced Wasserstein (DSW) [R.8] where the distribution over the aggregated projecting direction is not uniform and is not supported on the unit-hypersphere. Also, the distribution over the aggregated projecting direction is regularized implicitly by architectures while DSW uses a loss-regularizer. We would like to recall that the other benefit of using convolution is saving computation and memory.
> > > > >
> > > > >
> > > > > [R.8] Distributional sliced Wasserstein and Applications to Generative modeling; Nguyen et al

---

> ### Author Response · Authors · 2022-07-29
> **Response to Reviewer GwEg - Part 2**
>
> **Q2**: While this defined CSW was shown to be a pseudo-metric in Theorem 1, it may be a serious weakness as a loss to be used in the optimization. Will the different image with the distance 0 be good to represent Unfortunately, there is no discussion on this issue. The non-linear extension of CSW was mentioned in lines 328-332, it does not seem to be meaningful as a metric considering that the linear version of it is a pseudo-metric
>
> **A2**:  Thank you for your insightful comment. Our response to your concerns is three-fold.
>
> 1. The CSW is a pseudo metric on the space of all distribution over tensors, which means we do not assume any structure on distribution over images. In practice, many empirical investigations show that image datasets belong to some geometry group (symmetry, rotation invariant, translation invariant, and so on). Therefore, the set of distributions over images might be a subset of the set of distributions over tensors.  If the convolutional transform can hold the injectivity on the set of distributions over images, CSW can be a metric on the space of distributions over images. In our applications, we compare the value of sliced Wasserstein and convolution sliced Wasserstein on MNIST digits in Table 5 in Appendix F1, we found that the values of CSW are closed to the value of SW that can be considered as a test for our hypothesis of metricity of CSW. To our best knowledge, there is no formal definition of the space of distributions over images and its property. Therefore, we leave this for future work.
>
> 2. Moreover, in deep learning applications, sliced Wasserstein is computed between empirical distributions over mini-batches of samples that are randomly drawn from the original distribution [R.1]. This is known as mini-batch optimal transport with sliced Wasserstein kernel that is used when dealing with very large scale distributions and implicit continuous distributions. When using mini-batches, both Wasserstein distance, sliced Wasserstein distance, and convolutional sliced Wasserstein will lose its metricity to become a loss [R.1]. Therefore, metricity is not the only deciding factor in some applications of sliced Wasserstein, such as deep generative model, deep domain adaptation, and so on. This partially explains the better performance of CSW  on our deep generative model experiments in Table 1.
>
> 3. According to non-linear extension of CSW, we would like to mention that almost all deep learning applications use sliced Wasserstein on vectorized features of a non-linear convolutional neural network which can be considered as a more complex form of non-linear CSW (e.g, deep generative model [R.2],[R.3], deep domain adaptation [R.4]). Therefore, the definitions of CSW and non-linear CSW are the first building block to understanding practical approaches and are also the first attempt to incorporate geometry into sliced Wasserstein distance.
>
> --- Finally, we have added these discussions to the paper in Appendix A in the revision (in blue color).
>
> [R.1] Learning with minibatch Wasserstein : asymptotic and gradient properties; Fatras et al.
>
> [R.2] Generative Modeling using the Sliced Wasserstein Distance; Deshpande et al
>
> [R.3] Sliced Wasserstein Auto-Encoders; Kolouri  et al
>
> [R.4] Sliced Wasserstein Discrepancy for Unsupervised Domain Adaptation; Lee et al

---

> > ### Comment · Reviewer_GwEg · 2022-08-08
> > **Response to "Response to Reviewer GwEg - Part 2"**
> >
> > I would like to thank the authors for their responses on the issue of "pseudo-metric". I think this discussion is worth mentioning in the main text, not just in the Appendix.

---

> > > ### Author Response · Authors · 2022-08-08
> > > **Response to Reviewer GwEg**
> > >
> > > Thank you for your response.
> > >
> > > We will move the discussion into the main text in the revision.
> > >
> > > Best regards,

---

> ### Author Response · Authors · 2022-07-29
> **Response to Reviewer GwEg - Part 3**
>
> **Q3**: Compare to recent max SW or generalized sliced Wasserstein [23]... Comparing with recent Wasserstein variants and investigating proper (or optimal) convolution operators could improve this manuscript.
>
> **A3**: Thank you for your suggestions. We agree the max-convolutional sliced Wasserstein (Max-CSW) is a great extension. We define Max-SW and its variants in Definition 9 in Appendix E, and conduct experiments to compare it with max sliced Wasserstein on CIFAR10, CelebA, and CelebAHQ. The result is shown in Table 6 in Appendix F.2 in the revision. From the result, we observe that there is always a variant of Max-CSW that is better than Max-SW on CIFAR10, CelebA, and CelebAHQ. Moreover, we would like to mention that, similar to CSW, Max-CSW variants have fewer parameters and are faster than Max-SW. We can also adjust the configuration of the convolution slicer to get better performance. Moreover, we also generalize Max-CSW to convolution projected robust Wasserstein (CPRW) and its variants in Definition 11 in Appendix E. We observe that CPRW-stride is better than the projected robust Wasserstein on CIFAR10 based on the experimental result in Table 6 in Appendix F.2.
>
> According to the generalized sliced Wasserstein (GSW), using a circular defining function does not work in practice while the polynomial defining function cannot be computed in high-dimension. Therefore, in Table 6 (before revision) and Table 7 (after revision) in Appendix F.2, we have already compared non-linear CSW to GSW with the Sigmoid defining function. The table indicates that non-linear CSW is better than GSW. We would like to recall that all GSW variants [R.5] use projecting directions that are in the same dimension as the supports of distributions while Non-linear CSW can save memory and computation with convolution operators.
>
> [R.5] Generalized Sliced Wasserstein Distances; Kolouri  et al
>
> **Q4**: Moreover, there were many different convolutions defined in the main text and compared in the experiments, the results seem to suggest no clear winner among these convolutions. Thus, the experiments in their current form seem weak.
>
> **A4**: Overall, we observe that the convolution stride is better on almost all datasets such CelebA, CIFAR10, and CelebAHQ. Therefore, we recommend this variant. Moreover, we believe that the distributions on images of different datasets have different geometry structures. Therefore, a specific type of convolution might be preferred by a specific dataset that partially explains why some CSW-variants are better in some datasets while being worse in other datasets. Investigating geometric structures of data that are preferred by each variant of CSW is an interesting research direction and we will leave this direction to future work.
>
> **Q5**: Conventional image processing works have widely used vector representations in their models and tensors (i.e., reshapes) in their implementations. While this manuscript mentioned that "a probability measure over images should be defined over the space of tensors instead of images" and "this extra step does not keep the spatial structures of the supports, which are crucial information of images", I need more explanation on why these sentences are justified. Even though images are represented as vectors, spatial structures can be considered in the operations on them.
>
> **A5**: Thanks for your comments. Indeed, there is a typo. "A probability measure over images should be defined over the space of tensors instead of images" should be "a probability measure over images should be defined over the space of tensors instead of vectors". Thank you for pointing out this typo.
>
> As we clarified in A1, doing vectorization has two limitations which are destroying the geometry of images and requiring more memory and computation for projection. Also, doing reshaping/vectorization is a one-to-many mapping since we have different ways to arrange entries of tensors into vectors. Therefore, we can consider vectorization as a redundant operator. Moreover, preserving geometry and spatial structures when doing vectorization is hard. For example, Vision Transformer [R.6] needs to divide the images into 16x16 smaller images and then do vectorization. That is one example of the importance of spatial information on images. To our best knowledge, there is no simpler choice than using convolution directly on images to preserve the geometry.
>
> [R.6] An Image is Worth 16x16 Words: Transformers for Image Recognition at Scale; Dosovitskiy et al
>
> **Q6**: potential negative societal impact?
>
> **A6**: We leave the discussion on negative societal impact in Appendix A.  Overall, our proposed CSW is a general tool, hence, it could be used in applications that do not have a good purpose such as creating people's images without permission or attacking machine learning systems. We have also added a paragraph to discuss the limitations in Appendix A  in the revision.

---

> > ### Comment · Reviewer_GwEg · 2022-08-08
> > **Comments on "Response to Reviewer GwEg - Part 3"**
> >
> > I would like to thank the authors for their responses. For Q3, please see my comments on "Response to Reviewer GwEg - Part 1" and feel free to discuss further if possible to show clear differences between this work the work of [23].

---

> > > ### Author Response · Authors · 2022-08-08
> > > **Response to Reviewer GwEg**
> > >
> > > Thank you for your response.
> > >
> > > We have discussed the differences between our work and the work of [23]. We refer the reviewer to the corresponding discussion for detailed comparisons.
> > >
> > > Best regards,
> > >
> > > Authors,

---

> ### Author Response · Authors · 2022-08-09
> **Looking forward to your feedback.**
>
> Dear Reviewer GwEg,
>
> Given that we already addressed your concerns, the discussion deadline is only a few hours from now,  and you give a negative score on the paper, we would like to hear your feedback on whether our response is sufficient to change your opinion about the paper. Please feel free to raise questions if you have other concerns.
>
> Best regards,
>
> Authors

---

### Author Response · Authors · 2022-07-29
**Summary of the revision**

Dear Reviewers and Chairs,

We would like to thank the reviewers for their time and feedback. We have answered all questions of the reviewers in the corresponding discussions. Moreover, we have also included the following results (written in blue color) in our revision:

1. We introduce max convolution sliced Wasserstein (Max-CSW) variants which are extensions of max sliced Wasserstein with convolution slicers. Moreover, we also define convolution projected robust Wasserstein (CPRW) which is an extension of projected robust Wasserstein (PRW) (projecting measures into $k>1$ dimensional subspace). The definitions of Max-CSW and CPRW are given in Definition 9 and Definition 11 in  Appendix E. We compare Max-CSW variants to Max-SW on CIFAR10, CelebA, and CelebA-HQ datasets. The results are given in Table 6 in Appendix F.2. From the table, we observe that the Max-CSW-s variant gives the best result on CIFAR10 and CelebA while Max-CSW-d performs the best on CelebA-HQ. Moreover, we also compare the CPRW-stride variant to PRW in our revision. In particular,  we run experiments on CIFAR10 with the subspace dimension $k \in \{2,4,16\}$. For all choices of $k$, CPRW-stride has lower FID scores than PRW. The above results reinforce the claim that using convolution for projecting measures over images leads to more meaningful ways of comparing measures.

2. We have added the results of SW, CSW-b, CSW-s, CSW-d with (L=100, L=1000) on CIFAR10 after running 5 different times in Table 1 in the main text. From the table, we still observe the same phenomenon that CSW variants are better than SW in terms of FID scores and IS scores.

3. We have also submitted the code for the new experiments.

4. We have fixed typos and revised the writing based on the suggestions of reviewers in blue color. We have also added a paragraph for discussing the limitations of CSW and questions of reviewers in Appendix A.

We are looking forward to your feedback.

Best regards,

Authors

---

### Author Response · Authors · 2022-08-08
**Comparison to the neural network version of "Generalized sliced Wasserstein distance" and distributional sliced Wasserstein distance**

As in discussion with Reviewer GwEg and nLU5, we would like to clarify our contribution of CSW compared to the neural network version of generalized sliced Wasserstein [23 and distributional sliced Wasserstein [R.8] (DSW).

[23] Generalized sliced Wasserstein distances; Kolouri  et al

[R.8] Distributional sliced Wasserstein and Applications to Generative modeling; Nguyen et al

### **Comparision to the neural network version of generalized sliced Wasserstein**

1. The authors in [23] did not define the space of projection rigorously for the neural network which makes the study of sample complexity impossible. In contrast, we define rigorously the projection space of CSW which is the set of all convolution kernels that has the norm 2 square equals 1. This rigorous definition helps us to derive the sample complexity of CSW in Proposition 2.

2. The projection complexity of a general neural network-based projection is high. For example, the authors in [23] propose to use a multi-layer perception (MLPs) with Leaky-ReLu activation that leads to a very large number of parameters for the space projection which is the space of all possible values of the weights of the MLPs. Therefore, the computational complexity and projection complexity are huge compared to the conventional sliced Wasserstein. In contrast, our CSW has a lower computational complexity and projection complexity than the conventional SW due to the lightweight convolution kernel. In [23], the authors must use the max slice variant (Max-GSW-NN) for saving the memory, however, the authors conducted experiments only on gradient flow application which is not very large scale.  Moreover, the authors in [23] did not publish the code for that application https://github.com/kimiandj/gsw. In the deep generative modeling setting, the performance in terms of generative quality of Max-GSW-NN is only slightly better than SW while the computational time is much slower [R.8]. In contrast, CSW is better than SW while being faster.

[R.8] Distributional sliced Wasserstein and Applications to Generative modeling; Nguyen et al

3. Again, the most important contribution is that GSW-NN is still defined on **vector** space. It is the reason why the authors in [23] propose to use MLPs instead of convolution neural networks.  In contrast, we are the first people that define a sliced Wasserstein variant between distributions over **tensors** or images. We would like to recall that one of our contributions is changing the way of defining and interpreting sliced Wasserstein variants on different types of data.  It is not just about choosing the architecture of neural work.  The most crucial message that we want to bring in is incorporating geometry into the design of slicing operators for different types of data.

4. The generalized sliced Wasserstein defines with a general slicing function $f_\theta$. Hence, everything slicing function can be considered a special case of $f_\theta$. However, designing an efficient parameterization of $f_\theta$  is a challenge that has not been solved and that challenge has prevented the applications of GSW-NN in practice. In the paper, we make the first step to shedding light on the "black-box" $f_\theta$ and make it possible to be used in large-scale applications.

### **Comparison to distributional sliced Wasserstein [R.8] (DSW)**

As in discussion with Reviewer nLU5 , consecutive convolution operations can be replaced with a single convolution; however, the distribution of that “single convolution” can be complicated. Therefore, it is impossible to sample from that distribution.  To ease the ensuing discussion, we refer to the “single convolution” as the aggregated projecting direction variable. CSW can be seen as a special case of distributional sliced Wasserstein (DSW) [R.8] where the distribution over the aggregated projecting direction is not uniform and is not supported on the unit-hypersphere. Also, the distribution over the aggregated projecting direction is regularized implicitly by architectures while DSW uses a loss-regularizer. We would like to recall that the other benefit of using convolution is saving computation and memory.

[R.8] Distributional sliced Wasserstein and Applications to Generative modeling; Nguyen et al


Best regards,

Authors,

---

### Author Response · Authors · 2022-08-08
**Comparison between convolution and the equivalent Toeplitz matrix multiplication**

As in discussion with Reviewer GwEg, we would like to compare the benefit of using convolution directly on **tensor** space with the equivalent Toeplitz matrix multiplication on the **vector** space.

Theoretically, 2D convolution can be converted to matrix multiplication. However, that will create a lot of redundantly zero multiplications which worsen the computation complexity and the memory complexity. This is also an issue of the vectorization step due to defining the sliced Wasserstein variant on **vectors**. For example, a tensor of size $c \times d \times d$ ($d$ is even can be mapped to a tensor of size $1 \times d/2 \times d/2$ with the kernel of size $c\times 2 \times 2$ and stride $2$ (invariance to the dimension $d$) and with time complexity $\mathcal{O}\left(cd^2 \right)$. When using Toeplitz matrix multiplication, the size of the  Toeplitz matrix is $cd^2 \times d^2/4$ which leads to the time complexity of $\mathcal{O}\left(\frac{1}{4}cd^4 \right)$. We would like to mention that convolution can be implemented directly on the hardware [R.1,R.2,R.3].

[R.1] Accelerating Deep Convolutional Neural Networks Using Specialized Hardware, Ovtcharov et al.

[R.2] Hardware Accelerated Convolutional Neural Networks for Synthetic Vision Systems, Farabet et al.

[R.3] A Unified Hardware Architecture for Convolutions and Deconvolutions in CNN, Bai et al

Best regards,

Authors,

---

### Meta-Review · Area_Chair_x6rB · 2022-08-23

**Recommendation:** Accept
**Confidence:** Less certain

**Metareview:**

The paper presents a new slicing methods for the Wasserstein distance between probability measures over images based on convolution operators. This way memory requirements can be reduced and locality can be better preserved. Experiments are conducted on generative modeling problems.
Reviewers noted that the idea of convolution operators on probability measures over images is natural and simple, yet novel and acknowledged theoretical and practical results. The rebuttals were in-depth and provided additional clarifications.
On the other hand reviewers note that CSW only defines a pseudo-metric.
Overall this paper is an interesting contribution to the NeurIPS community and should be accepted.

**Award:**

No

---

### Decision · Program_Chairs · 2022-09-14

Accept